# Joint coding of shape and blur in area V4

Timothy D. Oleskiw[1,2], Amy Nowack[2] & Anitha Pasupathy[2]

Edge blur, a prevalent feature of natural images, is believed to facilitate multiple visual processes including segmentation and depth perception. Furthermore, image descriptions that explicitly combine blur and shape information provide complete representations of naturalistic scenes. Here we report the first demonstration of blur encoding in primate visual cortex: neurons in macaque V4 exhibit tuning for both object shape and boundary blur, with observed blur tuning not explained by potential confounds including stimulus size, intensity, or curvature. A descriptive model wherein blur selectivity is cast as a distinct neural process that modulates the gain of shape-selective V4 neurons explains observed data, supporting the hypothesis that shape and blur are fundamental features of a sufficient neural code for natural image representation in V4.

---

[1] Department of Applied Mathematics, University of Washington, Seattle, WA, USA. [2] Department of Biological Structure, University of Washington, Seattle, WA, USA. Correspondence and requests for materials should be addressed to T.D.O. (email: oleskiw@uw.edu)

In any natural scene, visual information is carried by boundaries of contrast that exist throughout an image[1]. For example, figure–ground contrast along the borders of solid objects may provide robust cues for object shape[2], while internal boundaries and surface texture may reveal 3D structure and material composition of those objects[3, 4]. Much work has quantified the extent to which these edge cues contribute to complex visual tasks, such as segmentation and recognition[5], and progress is being made toward understanding the neural mechanisms responsible (e.g., see refs. [4, 6–9]).

However, physical environments under naturalistic viewing conditions often produce edges that are blurred, i.e., exhibit a spatial gradient of image intensity across the edge (Fig. 1a, b). Specifically, edges without blur are sharp step transitions in intensity, whereas blurred edges vary smoothly in intensity from one side to the other. Such blurred boundaries within natural images can arise from a number of physical scenarios, such as defocus, cast shadows, or surface shading[10], and thus themselves convey relevant scene information such as object depth[11, 12]. Importantly, computational studies of luminance boundaries find that visual scenes may be sufficiently reconstructed from information contained in edge features, including the magnitude of blur at each edge[10]. Further, psychophysical results demonstrate that in addition to shape[13], the visual system is tuned to detect cast shadows during segmentation[14], of which shape and blur are diagnostic features.

The human visual system is also adept at discriminating blur[15] and detecting blurred boundaries[14, 16, 17]. While biophysically plausible computational models have been proposed to explain how the brain could utilize blur information[18], neural mechanisms that underlie the computation and representation of blur remain unclear. Since sharp and blurred boundaries differ greatly in their high spatial frequency (SF) content, V1 populations tuned to various SFs implicitly encode blur. However, at intermediate stages of form processing, such as in area V4, simple gratings are ineffective at driving responses of shape-selective neurons[19], and complex shape stimuli that do elicit responses have typically been defined by sharp boundaries[20–22]. Thus, it is unknown whether and how blur is encoded and combined with shape information along the ventral pathway to form a sufficient representation of natural scenes.

Here we present results from a study targeting single V4 neurons using customized sets of shape stimuli to test the hypothesis that V4 neurons jointly encode object shape and boundary blur. Our results demonstrate that shape-selective V4 neurons also exhibit tuning for blur and that single-unit responses are well described by a joint model explicitly encoding both shape and blur information.

## Results

**Selectivity for blur in area V4.** To understand how blur, i.e., the gradient of image intensity, is encoded in the intermediate stages of the ventral pathway, we examined the responses of well-isolated V4 neurons to shape stimuli as a function of blur magnitude. For each neuron, we first assessed shape selectivity using a standard stimulus set (Fig. 1c, d; ref. [22]). Based on these responses, we identified a subset of preferred and non-preferred stimuli that evoked a range of responses for the neuron in question (see Methods: 'Visual stimulation'). We then examined the responses to this chosen subset of stimuli under various levels of blur (Fig. 1e).

Blurring a stimulus boundary, as implemented here (see Methods: 'Visual stimulation'), broadens the intensity gradient across a shape's boundary. Because the responses of roughly 80% of V4 neurons increase as figure–ground stimulus contrast is increased[23], one may expect blur to reduce the response of shape-selective neurons as edge intensity gradients are broadened. Indeed, many neurons in our population follow this trend. For example, cell a23 in Fig. 2a exhibited a range of responses from 15–45 spk/s for a variety of sharp stimuli subjected to minimal blur ($\beta = 0.005$). This response pattern was maintained for small amounts of blur ($\beta < 0.04$), but for intermediate and high blur factors ($\beta \geq 0.04$) responses to preferred stimuli gradually declined; at the highest levels of blur tested, i.e., $\beta = 0.64$, all stimuli were effectively amorphous with little discernible form (Fig. 1e), and responses approach baseline (dashed line). Thus, for this neuron both response magnitude and shape selectivity declined with increasing levels of blur. Figure 2b, c illustrate additional examples of this general behavior, but rather than a gradual decline as in Fig. 2a, cells a08 and b29 of Fig. 2b, c maintained shape selectivity up to blur factor $\beta = 0.16$ before transitioning sharply to a baseline level that is not selective for

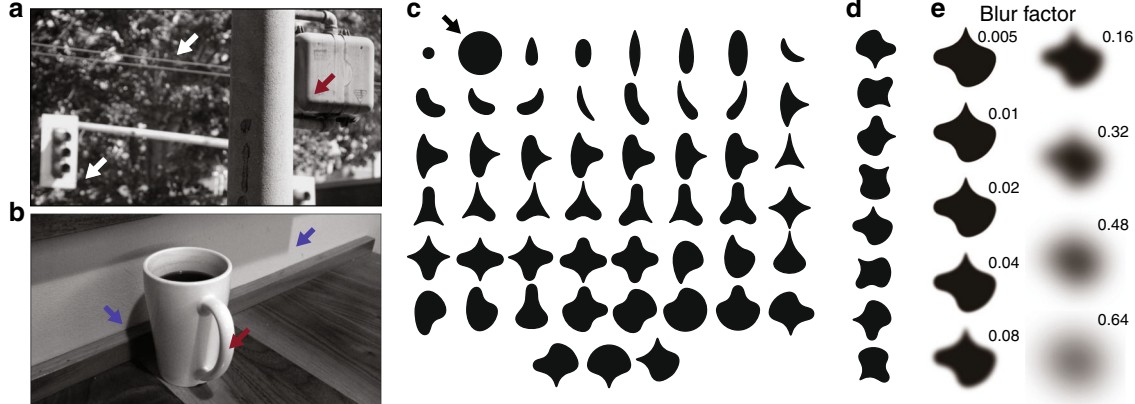

**Fig. 1** Examples of blur in natural images and stimuli used to explore selectivity for shape and blur. **a, b** Examples of different types of blur in natural scenes. **a** Focal blur (white arrows) conveys information about depth while shading blur (red arrows) conveys information about 3D structure. **b** Penumbral blur is associated with cast shadows (blue arrows); during grouping, cast shadows do not interfere with perception of physical object boundaries and shading. **c–e** Stimulus set used to assess tuning for shape and blur in V4 neurons. **c** A standard set of 51 shapes were used to assess shape selectivity of V4 neurons. Stimulus size is defined relative to the diameter of the large circle (black arrow). **d** Each shape was presented at up to 8 unique orientations at 45° increments; all rotations for one example shape are shown. For shapes with radial symmetry, duplicates were excluded. **e** To assess tuning for blur, a subset of preferred and non-preferred shapes were presented at up to 9 levels of Gaussian blur (see Methods: 'Visual stimulation'). Example stimuli $\beta \geq 0.32$ were cropped here for display purposes

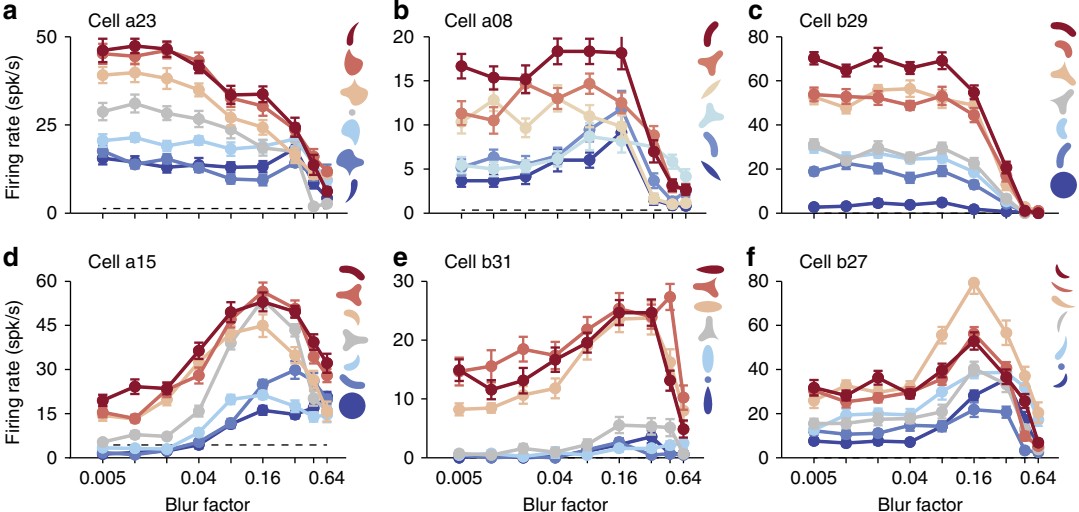

**Fig. 2** Shape-selective V4 neurons are tuned for blur. **a–f** For each neuron we plot the mean responses ($y$-axis) to several stimuli as a function of the magnitude of blur factor ($x$-axis, $\beta$). Line color indicates shape identity and is ordered from preferred (red) to non-preferred (blue) stimuli for each neuron based on responses to the sharp versions of each stimulus ($\beta = 0.005$). Error bars indicate s.e.m. **a** Responses of an example V4 neuron that was strongly selective to sharp stimuli, i.e., $\beta = 0.005$; responses declined gradually to baseline levels (dashed line) as blur magnitude was increased. **b, c** Two additional examples that also exhibited a monotonic decrease in responses with increasing blur. Unlike **a**, these neurons maintained their response level across low blur levels, sharply declining to baseline beyond a critical blur factor ($\beta \approx 0.16$). **d–f** Example V4 neurons that respond best at intermediate levels of blur; responses for preferred stimuli dramatically increase for intermediate blur factors

shape. Note that limited shape selectivity may occur at high blur ($\beta \geq 0.32$) since all stimuli retain low SF-oriented energy as blur magnitude is increased.

In striking contrast, many other V4 neurons exhibited a marked increase in response magnitude over intermediate blur levels (Fig. 2d–f). In other words, the activity of these three example neurons was non-monotonic as a function of blur. Further, this blur modulation appears to depend on stimulus shape, facilitating responses of preferred shapes more than non-preferred shapes. As a result, in these neurons shape selectivity is strongest over intermediate blur factors. For all three examples, responses to blurred stimuli are highest for blur values between $\beta = 0.16$ and $\beta = 0.32$, overlapping with the range of blur values associated with a decline in responses seen in Fig. 2a–c.

To quantify the effect of blur across the population of V4 neurons, we performed a model-free analysis of blur modulation illustrated in Fig. 3. For each neuron we first constructed an average tuning curve as a function of boundary blur, based on the interpolated responses to a subset of preferred stimuli (see Methods: 'Analysis and model fitting'). We then calculated two metrics from each tuning curve: the extremal blur factor (Fig. 3a–d, triangles) that is associated with the maximal response modulation relative to average activity evoked by non-blurred (sharp) stimuli, and a modulation index taken as the tuning curve integral across blur factors (Fig. 3a–d, hatching). Figure 3e depicts the modulation index (MI, $y$-axis) as a function of extremal blur factor ($\beta$, $x$-axis) for all neurons in our population ($n = 65$); in this space our recorded data span a continuum, and we demarcate cells via modulation index and extremal blur factor criteria. Immediately visible is a sub-population ($n = 42$, $\approx 65\%$) with a high extremal blur factor and negative modulation index (MI < 0, $\beta > 0.32$); these neurons exhibit responses that decrease with increasing blur, collapsing to a near-baseline response at highest blur levels (e.g., cell a19 of Fig. 3d). Note also that a few cells ($n = 5$, $\approx 8\%$) have weak tuning for intermediate blur coupled with a strong fall-off at high blur values (e.g., cell b13 of Fig. 3b) to produce a non-negative modulation index at high extremal blur factors (MI $\geq 0$, $\beta > 0.32$). Conversely, other neurons ($n = 11$,

$\approx 17\%$) exhibit a non-negative modulation index with intermediate extremal blur factor values (MI $\geq 0$, $0.1 < \beta < 0.32$), indicative of neurons tuned to intermediate blur magnitudes (e.g., cell a15 of Fig. 3a). Interestingly, some cells ($n = 7$, $\approx 10\%$) demonstrate intermediate inhibition, i.e., negative modulation at intermediate blur values (MI < 0, $0.1 < \beta < 0.32$); these neurons exhibit strong shape selectivity at both low and high blur magnitudes. In Fig. 3e the first principal component of our population in this space, calculated under a scaling to equalize variance along each dimension (shaded line), aids in segregating our neurons: cells with a positive principal value (PV, more red) respond best to intermediate blur, while those with a negative PV (more blue) show declining responses with increasing blur. While it is not perfect, we see in Fig. 3f that by superimposing blur tuning curves across the population, colored according to each neuron's PV, this measure is diagnostic of selectivity for low blur (more blue) and intermediate blur (more red).

**Controlling for stimulus size**. In addition to the gradient width of edge intensity, blurring alters many other stimulus characteristics. For example, as depicted in Fig. 4a, the foreground area of a blurred shape stimulus, defined as the number of pixels distinct from the background, increases with blur magnitude. Thus, tuning for boundary blur might arise from a simple tuning for stimulus size. To test this hypothesis, in a subsequent size control experiment, we presented shape stimuli that were first scaled by $\pm 10\%$ and then subjected to a diagnostic subset of blur levels (Fig. 4b). If preference for intermediate blur was simply due to a preference for stimulus size we would expect a shift in the blur tuning peak as size was varied. In Fig. 4c, d we plot the responses of two example neurons that respond preferentially to intermediate levels of blur. For both examples the modulation of responses with respect to blur was consistent across changes in stimulus size. Across the neurons subjected to this size control ($n = 26$), in every case we found that blur modulation was similar across stimulus size. Importantly, while one might expect a systematic variation in responses across all stimuli with respect to

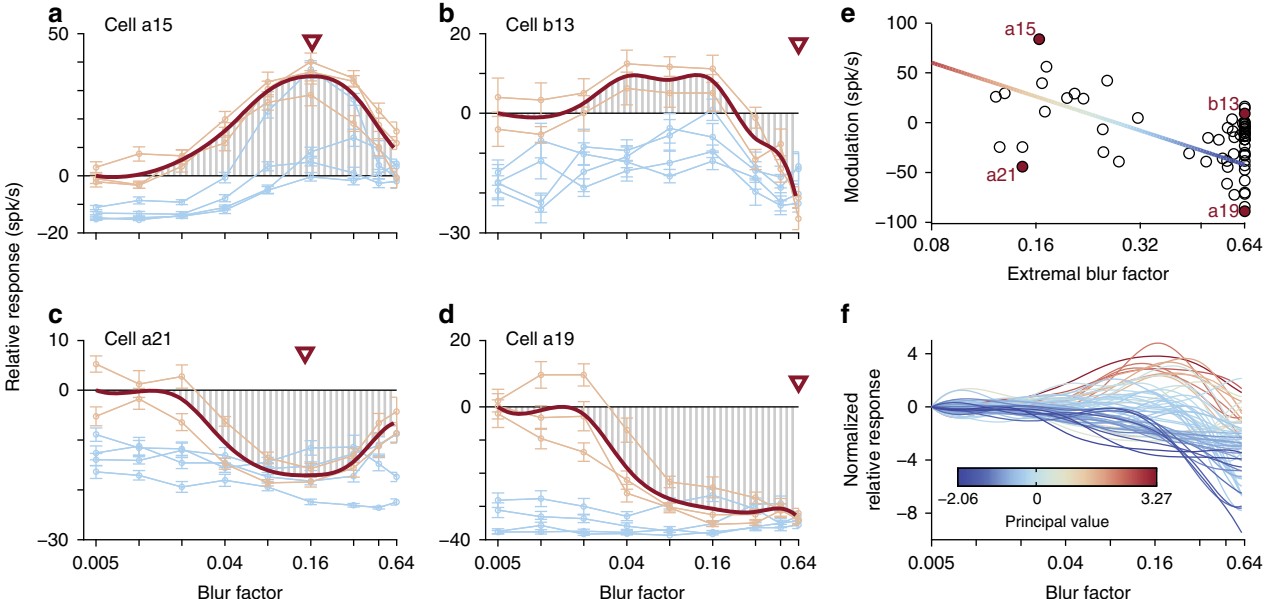

**Fig. 3** Model-free analysis of blur selectivity across cells. **a–d** Average blur tuning curves of four example neurons (dark red) constructed by averaging responses to preferred shape stimuli (light red). Relative response (y-axis) as a function of blur factor (x-axis, β) was computed with respect to mean response across sharp preferred stimuli, i.e., relative response is zero for lowest blur factors ($\beta = 0.005$). An extremal blur factor (triangle) was defined as the magnitude of blur that evoked the largest absolute deviation relative to responses to sharp stimuli. Response modulation was determined by calculating the integral of relative responses across blur factors (hatching). **e** Response modulation (y-axis) is plotted as a function of extremal blur factor (x-axis) for the population of neurons ($n = 65$) in our data set. The principal value, calculated from the first principal component of the population (shaded line), demarcates neurons with peak responses at intermediate blur values (more red) from those that show declining activity as blur increases (more blue). Example cells of **a–d** are filled and labeled. **f** Superposition of blur tuning curves computed in **a–d**, scaled to have a unit-variance of relative response (y-axis) and colored according to **e**, demonstrate complementary tuning with respect to blur magnitude (x-axis) within our population

size, we did not find a significant interaction effect between size and blur for any of the neurons (two-way ANOVA, $p > 0.14$ for all cells, median $\approx 0.96$). Rather, blur accounted for a significant fraction of variance ($p < 0.05$) in the majority of these neurons ($n = 23$, $\approx 88\%$). As a result, our analysis suggests that selectivity for blur cannot be explained in the context of overall stimulus size. These findings were confirmed by separate analyses in which models of V4 shape selectivity were used to predict size control data. Briefly, while V4 responses to shape stimuli were adequately explained by existing models of boundary conformation[22], these models failed to predict responses to scaled and blurred shape stimuli (see Methods: 'Control experiments').

**Controlling for stimulus contrast**. A further confound arising from blur is the stimulus intensity contained within a shape's boundary. If we simply define the boundary contour of a blurred shape as the level set of stimulus intensities distinct from the background, we note from Fig. 1e that the average intensity within that boundary decreases as blur magnitude increases. That is to say, blur diffuses stimulus intensity, reducing the average contrast between figure and ground. Therefore, the preference for an intermediate level of blur could arise from a preference for a specific average stimulus intensity which differs from that of sharp stimuli.

To test this hypothesis, for each blurred stimulus we constructed a non-blurred (sharp) version matched in mean intensity (Fig. 5a, b) and compared the responses to these two stimulus sets. Here, mean stimulus intensity is determined from the area subtended by pixels differing from background intensity on our 24-bit color display. Figure 5c, d plots the results for two example neurons. For cell b26, the tuning curves as a function of blur and contrast are dramatically different: responses are strongest for intermediate blur, but fall off as contrast is reduced

for sharp stimuli, inconsistent with blur tuning explained by contrast. On the other hand, cell b32 demonstrates very similar selectivity across both stimulus sets, suggesting that blur selectivity in this case could be explained by a simple figure–ground contrast preference. To quantitatively compare the two tuning curves, we calculated the center-of-mass (CoM) for each: tuning curves that peak at intermediate contrasts will garner a large CoM, and curves which monotonically decrease will retain smaller CoM values. Here, bootstrapping is employed to estimate the distribution of tuning curve CoM measurements under the variance of observed responses (see Methods: 'Analysis and model fitting'). Across our sub-population of neurons subjected to the intensity control (Fig. 5e, 34 cells), the majority ($n = 31$, $\approx 91\%$) had an intensity-matched CoM significantly different than that of blur (t-test, $p < 0.05$), indicating neural responses are not consistent with tuning for intermediate stimulus intensity. Thus, while selectivity for blur could be attributed in some neurons to a simple tuning for intermediate edge contrast, e.g., cell b32, the majority of cells cannot be explained in this context.

**Controlling for curvature modification**. A more subtle confound of blurred stimuli arises when considering exactly how to define object shape in the presence of blur. Specifically, a blurred shape can be associated with any of a family of closed contours, each defined from the level set of stimulus intensity (Fig. 6b) and the magnitude of boundary curvature along each threshold contour may change as a function of blur. This confound between curvature and blur is significant from the perspective of shape coding, as previous studies have leveraged the manner in which smoothed boundary contours devolve into an ellipse to represent object shape[24]. Therefore, since the responses of many V4 neurons to shape stimuli can be explained in the context of tuning for

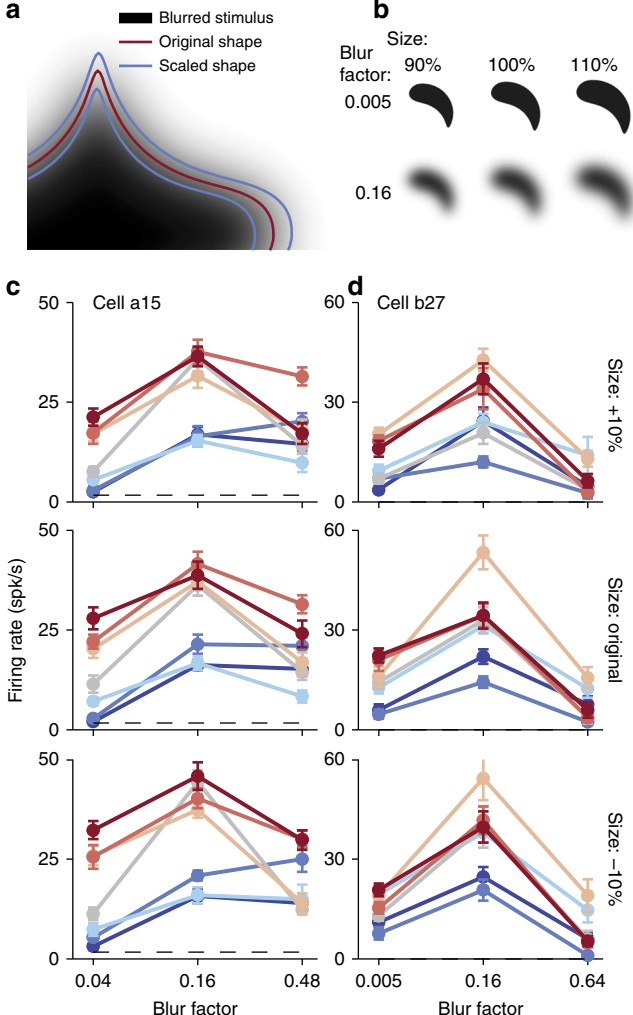

**Fig. 4** Stimulus size does not explain blur selectivity. **a** A blurred stimulus ($\beta = 0.16$) has an increased foreground area, defined as the number of pixels distinct from the background, compared to its original boundary prior to blurring (red). Stimuli scaled by $\pm 10\%$ are shown for comparison (blue), and correspond to luminance thresholds approximately 1/3 and 2/3 of maximum, respectively. **b** Example scaled and blurred stimuli used to assess a potential size confound. **c, d** Results of the size control experiment for cells a15 and b27 demonstrate increased responses for intermediate blur irrespective of stimulus size. Line color represents stimulus identity per Fig. 2d, f. For both neurons, responses were not significantly influenced by size ($p = 0.45$ and $p = 0.18$, respectively) but significant variance was found with respect to blur ($p < 0.0001$). Error bars indicate s.e.m.

boundary curvature[22], we wondered whether the selectivity for intermediate blur may be due to a preference for modified curvature values that arise in blurred stimuli that were not a part of the original sharp stimuli.

To test the hypothesis that blur selectivity is an epiphenomenon of shape selectivity we first described each neuron's shape preference in terms of tuning for boundary curvature. As done elsewhere[25–27], we identified the 2D Gaussian function in a shape space spanned by angular position and curvature (APC) that best predicts responses to the preliminary shape screen conducted using sharp stimuli (see Methods: 'Visual stimulation'). Bootstrapping was used to calculate the normalized mean-squared prediction error (Training NRMSE, Fig. 6d), as a measure of goodness of fit (see Methods: 'Analysis and model fitting'). We then evaluated how well this best-fitting APC model could predict

responses to blurred stimuli by considering the curvature descriptions associated with each blurred stimulus at a range of intensity thresholds (see Fig. 6a for a schematic of this procedure). For each intensity threshold, we quantified goodness of fit as the normalized root mean-squared error (NRMSE) between the predicted and observed responses. Then, for each neuron, the intensity threshold that minimized NRMSE was selected as the exemplar threshold (Threshold Curvature NRMSE, Fig. 6c, d). In Fig. 6c for each neuron we compare this threshold NRMSE against the NRMSE of a mean model that is agnostic to blur, i.e., a model that predicts identical responses to sharp and blurred versions of the same shapes. For most neurons in our population, model predictions derived from optimized intensity thresholds were associated with larger errors than simple predictions equal to mean responses across blur. Furthermore, Fig. 6d demonstrates that the APC model's failure to explain blur response variance is not due to an inability of the model to capture shape selectivity exhibited in our population; we find that, for the majority of neurons, prediction error of responses to blurred stimuli were higher than prediction error of models trained and validated on sharp stimuli alone.

**Joint coding of shape and blur**. Results thus far demonstrate that neuronal responses in V4 are modulated by boundary blur, and this modulation cannot be explained on the basis of tuning for size, contrast, or curvature. Therefore, our findings support the hypothesis of an underlying neural code for object shape and boundary blur, e.g., a representation of both boundary conformation and spatial gradient of edge intensity in the rate responses of single V4 neurons.

To rigorously test this hypothesis, we evaluate whether a joint model of shape and blur performs significantly better at predicting V4 responses than a marginal model that is tuned for shape alone. Importantly, this analysis builds on the well-studied APC model[22, 25–27] that is known to capture tuning for boundary conformation in shape-selective V4 neurons (see Methods: 'Analysis and model fitting'). For each cell we first fit a standard APC model to blurred responses under the assumption that neurons are invariant to boundary blur of shape stimuli. Thus, the APC model fits to the mean response of each shape across blur factors. We then augment the APC model to include a blur-selective term, taken to be log-normal in blur factor $\beta$. Simply put, responses $R$ based on the angular position, curvature, and blur (APCB) model are predicted via $R = APC \times B$, where the APC model, a shape-selective function of angular position and curvature, is multiplicatively scaled by $B$, a blur-selective function (see Methods: 'Analysis and model fitting'). Note that APC and APCB models were fit to the blurred data without including the preliminary screen data to ensure that any fit differences were not due to the number and diversity of stimuli.

In Fig. 7a, we plot accuracy as determined by prediction error (NRMSE) across all trained stimuli and responses for shape-coding models with (APCB, $x$-axis) and without (APC, $y$-axis) the influence of blur. Leave-one-out cross-validation demonstrates (see Fig. 7b, d, g, Methods: 'Analysis and model fitting') an increased NRMSE prediction error of the APC model relative to the APCB model (APC—APCB error). Thus, the APCB model better captures the behavior of neurons to blurred stimuli, and this increased performance is not due to overfitting of additional parameters. In the majority of neurons, inclusion of blur information significantly improved prediction performance to stimuli in our data set ($p < 0.05$ for $n = 53$ of 65, $\approx 82\%$), including neurons selective for intermediate blur (blur selectivity PV $> 0.25$, $n = 12$ of 15, Fig. 7g). Examples in Fig. 7c, e, h illustrate the effectiveness of the APCB model: it accurately captures a

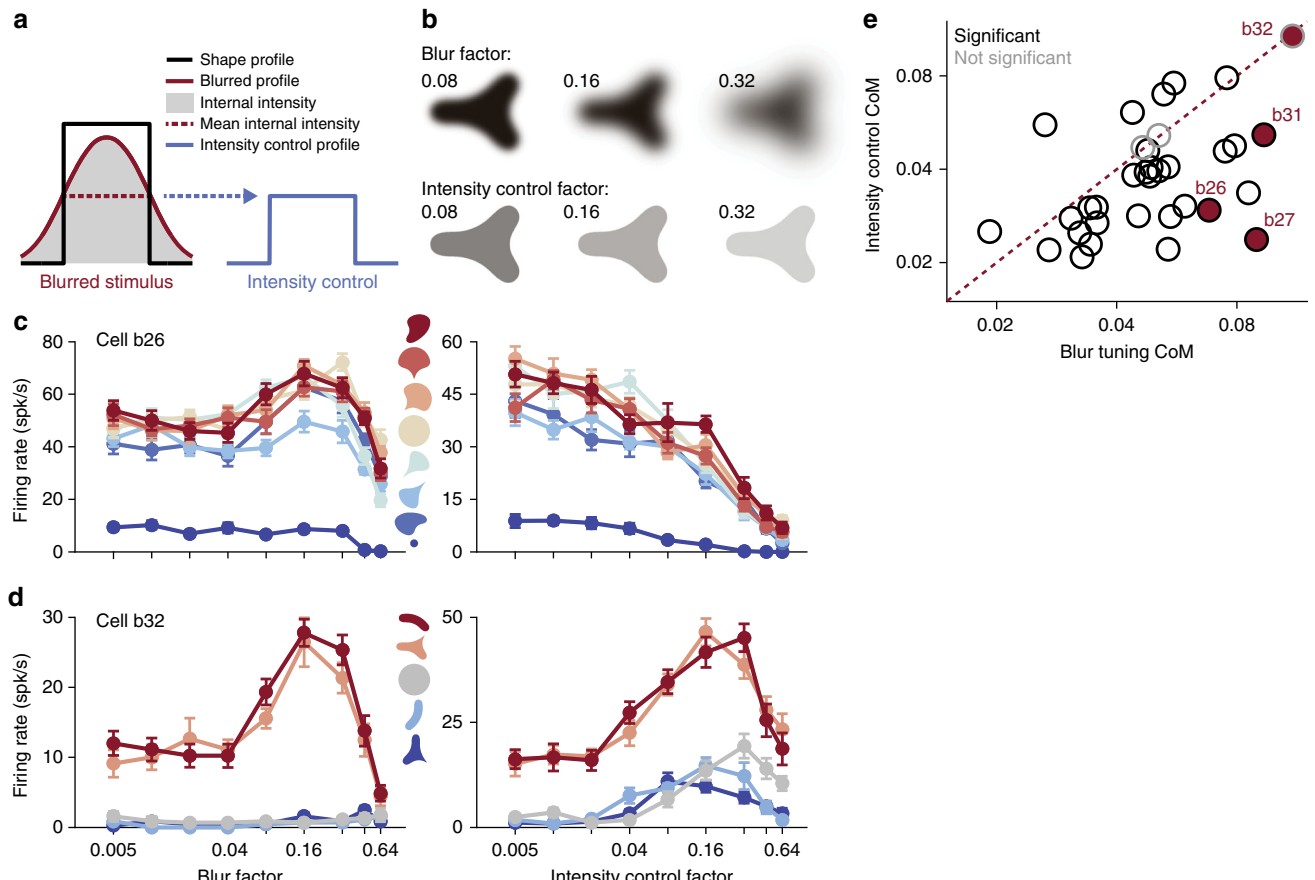

**Fig. 5** Stimulus contrast does not explain blur selectivity. **a** Profile schematic of how a blurred stimulus (red) has a decreased mean foreground intensity relative to a sharp ($\beta = 0.005$) stimulus (black). Low/high values correspond to background/foreground image intensities, respectively. An intensity control is constructed from a sharp shape with an identical mean foreground intensity (blue). **b** Example blurred stimulus and intensity-matched controls. Stimuli are shown in black but were presented at either positive or negative luminance contrast. **c**, **d** Responses of preferred (red) to non-preferred (blue) shapes that were presented either blurred (left) or as intensity-matched controls (right). While blur and contrast control tuning curves are remarkably different in **c**, they are quite similar in **d**. **e** Center-of-Mass analysis (see Methods: 'Analysis and model fitting') reveals that for a majority of cells ($n = 31$ of 34) blur and contrast-control tuning curves have significantly different (black) tuning profiles (t-test, $p < 0.05$); other cells (gray) exhibited blur and contrast-control tuning curves with CoM values that were not significantly different. For these neurons, blur tuning may be explained in the context of intensity tuning. Neurons above the diagonal typically exhibited a tuning preference for intermediate intensities while remaining largely invariant to all but the highest levels of blur; a more conservative estimate that discounts these cells ($n = 6$, $\approx 18\%$) still finds the majority of neurons unexplained by tuning for stimulus contrast ($n = 25$, $\approx 74\%$)

range of response behaviors including selectivity for intermediate blur (Fig. 7h) and fall-off of responses to baseline at high blur levels (Fig. 7c, e).

This choice of model parameterization, wherein blur tuning multiplicatively scales selectivity for boundary curvature, is but one of many possible descriptive models. For example, another formulation could have a selectivity for blur that additively facilitates position and curvature tuning, i.e., APC($\Gamma$;$\Theta$) + $B(\Gamma$;$\Theta$). In Fig. 7f, however, we find that the multiplicative APCB model outperforms the additive variant in the majority of neurons. Thus, we conclude that neurons in V4 jointly encode information of shape and blur, and that this coding is best explained by a gain modulation, with tuning for blur multiplicatively scaling shape selectivity. It is important to note that, by construction, the APCB model is separable, i.e., does not rely on the interaction between boundary conformation and blur to accurately predict neural response. As will now be addressed, this suggests a distinct neural mechanism regulating blur selectivity in V4.

**Distinct dynamic properties of blur-selective responses**. Finally, we ask how and when tuning for blur emerges in the the the

responses of individual neurons. In Fig. 8a, we illustrate, for the preferred shape of an example neuron (cell a23), the peristimulus time histograms (PSTH) as blur is varied. For this neuron responses to blurred shape stimuli decreased with increasing blur (Fig. 2a) and this decrease was uniform across the stimulus presentation interval. Given that shape and blur tuning is best explained by a modulation of gain, in Fig. 8c we quantify blur modulation by plotting the s.d. of responses across blur factors for both preferred (black) and non-preferred (gray) shape PSTHs. The difference between these curves (hatching) then captures the timecourse over which blur modulation is applied to shape-selective activity. The utility of this analysis is seen when considering a blur-selective neuron (cell a15) in Fig. 8b. Here, the PSTH of preferred shape stimuli demonstrates a nonuniform increase in responses, most pronounced over the initial (transient) wave of activity (approximately 50–150 ms after stimulus onset). This effect is captured by the transient response modulation seen in Fig. 8d. Thus, these example neurons suggest a potential difference in dynamics between two groups of cells: those which are selective for intermediate blur factors, and those which are not. If we demarcate neurons in our population with a

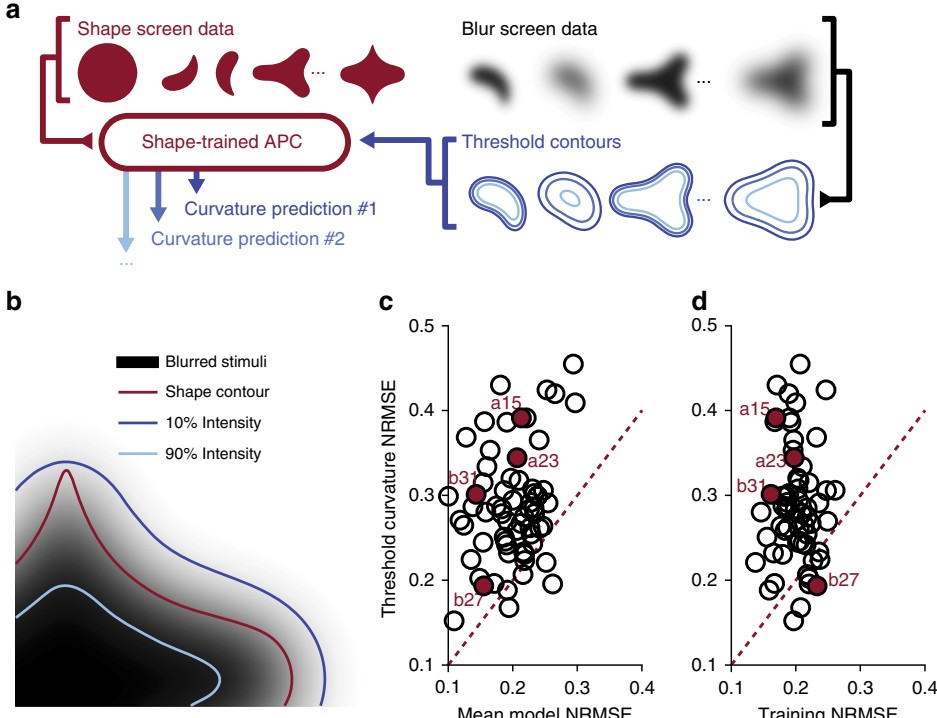

**Fig. 6** Curvature modification does not explain blur selectivity. **a** Schematic of analysis performed across shape and blur datasets to assess the contribution of curvature modification toward blur selectivity. An APC model is fit to data collected from shape screening (red), which then predicts responses to modified curvature threshold contours computed from blurred stimuli at different thresholds (blue). **b** A blurred stimulus ($\beta = 0.32$) generated from a shape contour (red) and a family of closed contours defined by the level set of an intensity threshold (blue), each with reduced curvature magnitudes. **c** For each cell, the minimum prediction error across all intensity thresholds (threshold curvature NRMSE; see Results) plotted as a function of a blur-invariant mean model's prediction error. The latter predicts responses to different shapes in accordance with the APC model ignoring blur; responses are identical overall blur levels for a given shape. Example cells are filled and labeled. **d** Threshold prediction error as a function of bootstrapped training error, i.e., a baseline estimate at how well the APC model predicts shape data

blur-selective principal value (Fig. 3e, f, shading) PV > 0.25 as tuned for intermediate blur from those tuned for sharp (minimal blur) stimuli, the average normalized PSTHs for intermediate-selective and sharp-selective sub-populations (Fig. 8e, f) exhibit the distinct qualitative differences observed in cells a23 (Fig. 8a) and a15 (Fig. 8b). For the blur-selective sub-population (Fig. 8f) differences between responses with respect to blur are transient, restricted to the early time period, while such differences exhibited by the sharp-selective sub-population (Fig. 8e) are uniform.

To quantify these differences we first plot in Fig. 8g blur-selective PV versus the difference of blur modulation between preferred and non-preferred shape stimuli for each cell (Fig. 8c, d, hatching) over the sustained period of activity (shaded, 200−300 ms). We note a significant anti-correlation between these quantities (Pearson's $r = -0.492$, $p = 0.052$); for cells with a larger PV, sustained modulation of gain is smaller. There was also a significant difference in sustained blur modulation between the intermediate-selective and sharp-selective sub-populations ($t$-test, $p = 0.018$). Further, we found no significant difference in shape-dependent blur modulation during the initial period of activation between the two sub-populations ($t$-test, $p = 0.154$), suggesting that the more selective for intermediate blur a cell is (high blur-selective PV), the more transient blur modulation appears to be (low average-sustained blur modulation).

While the response latency of cell a23 was unaffected by blur, cell a15 exhibited a consistent shift in tuning peaks: responses at high blur are significantly delayed. To examine this effect of response latency across the population for each neuron, we quantified the half-rise time as the duration between stimulus onset and half-of-maximum response at each blur factor. At low

blur factors ($\beta \le 0.16$), blur-selective cells tend to have a slightly (though not significant) shorter mean half-rise time on average (Fig. 8h). However, half-rise time increased significantly ($t$-test, $p = 0.036$) for blur-selective cells as the magnitude of blur increased (arrows). To factor out increased response latency as a population artifact, we analyzed latency differences based on the ratio of half-rise times as a function of blur factor, i.e., response latency normalized by each neuron's baseline half-rise time to sharp stimuli ($\beta = 0.005$), and find the same trend to hold (not shown). This suggests the underlying circuitry of V4 neurons tuned for intermediate blur is distinct from those which consistently reduce activity as blur increases; blur-selective neurons receive a transient multiplicative facilitation to shape-selective response as high-SF content is removed.

## Discussion

We studied the responses of primate V4 neurons to determine whether, and to what extent, blur influences neuronal activity. We found that many V4 neurons are jointly tuned to shape and blur, with responses explained by a blur-dependent gain-modulation of shape tuning. Importantly, our results are not a simple by-product of changes in stimulus size, contrast, and boundary curvature that co-vary with blur. Further, blur-dependent modulation of responses does not appear to be a strictly local mechanism, as modulation is consistent across multiple shapes with features presented in various RF subregions. This finding is reinforced by our size and curvature control experiments which demonstrate blur tuning to hold over local stimulus perturbations. Our demonstration of blur tuning implicates a role for V4

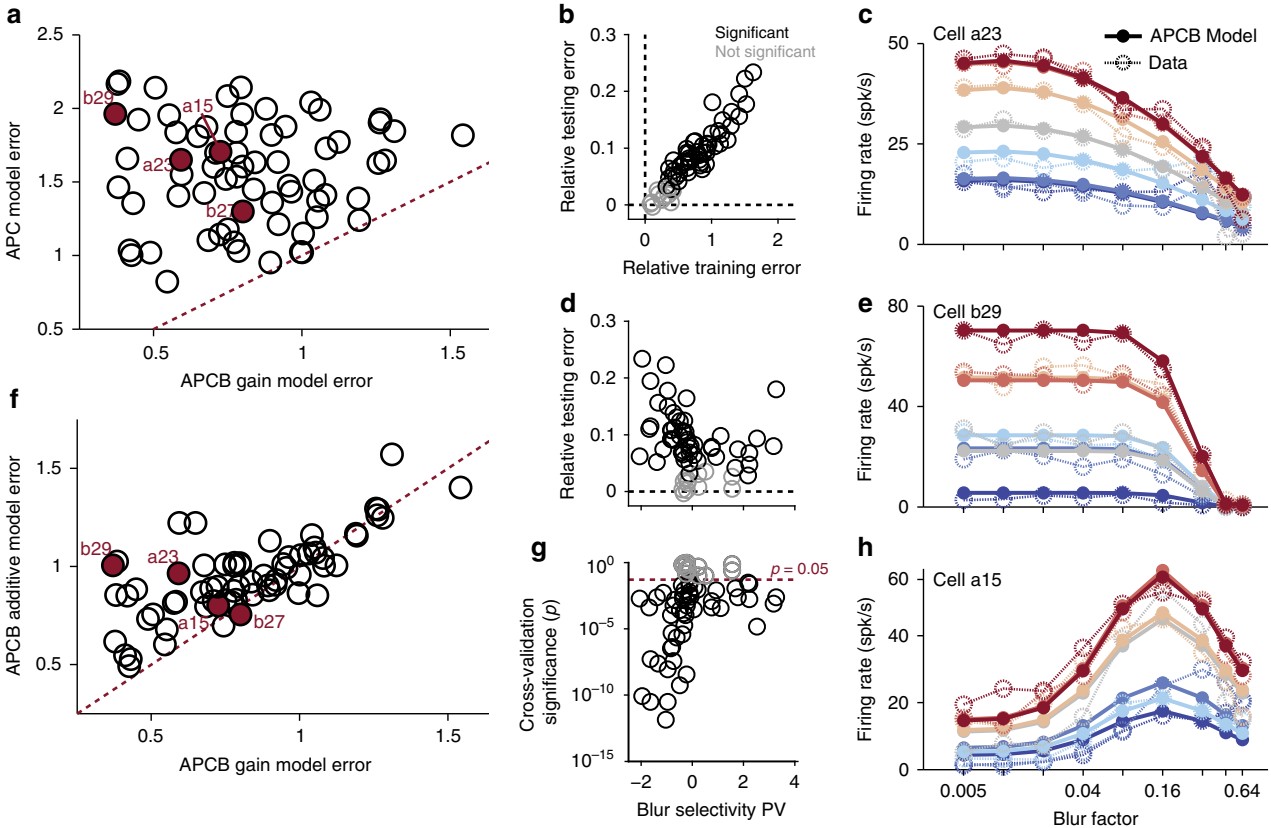

**Fig. 7** Responses explained by a joint model of shape and blur. **a** NRMS fit error of an APC model plotted as a function of NRMS fit error of an APCB model for each neuron. Example cells are filled and labeled. **b**, **d**, **g** Even though the APCB is a generalization of the APC model, i.e., APCB models form a superset of APC models with two additional parameters, cross-validation demonstrates the APCB model to better predict responses to blurred stimuli: **b** Average NRMSE across training and testing stimuli reveals the APCB model to better predict responses without overfitting; relative error, computed as APC—APCB error, is positive in all significant cases (**g**). **d** Relative testing NRMSE between models, plotted as a function of blur selectivity PV, demonstrates the APCB model to better predict responses over a range of neuron tuning profiles. **g** Significant (black) and not significant (gray) cases of prediction improvement were determined with a pair-wise *t*-test across hold-out validation stimuli (see Methods: 'Analysis and model fitting'). **c**, **e**, **h** Observed responses (open circles, dashed lines) and APCB model fits (filled circles, solid lines) for three example neurons (Fig. 2a, c, d). Qualitative assessment of fits suggests that the APCB model captures blur-tuned response properties remarkably well. **f** Comparison of the APCB gain model against an APCB additive variant with equal degrees of freedom (see Methods: 'Analysis and model fitting'). Most neurons are better fit by the APCB gain model, particularly when error is small, consistent with blur selectivity being explained by gain modulation

in processes cued by blurred boundaries, i.e., segmentation, depth perception, and shading, and supports the hypothesis that V4 can provide an explicit and sufficient representation of natural scenes.

Our results identify blur as a novel tuning dimension in visual cortex; while some V4 neurons exhibit a monotonic decline in shape-selective responses with increasing levels of blur, others maintain shape selectivity over a wide range of blur values, i.e., Fig. 2b, c. This latter group of neurons may be robust against the many physical scenarios in which contours may be blurred in naturalistic scenes. A separate group of neurons show maximal responses at intermediate blur levels, responding best when high-SF content is removed from a stimulus while preserving lower-band content. This effect cannot be explained by simple mid-band SF tuning, but rather indicates a preference for intermediate blur that requires high-SF information to suppress shape-selective responses. In V4, intermediate blur tuning is associated with response peaks between blur factors $\beta = 0.08$ and 0.16. At 3° eccentricity, for example, this corresponds to a response enhancement to frequency content between 2.6 and 1.2 cyc/°, respectively, which is consistent with SF tuning distributions observed in macaque V1[28]. Given that V4 responses are explained by a joint model of shape and blur, where shape-selective

responses are multiplicatively scaled as a function of boundary blur, blur tuning in V4 may arise from the aggregation of SF information reported by V1, consistent with previous demonstrations of V4 selectivity for non-Cartesian gratings[19] and illumination vectors[6]. Furthermore, tuned responses to shape occur at either low or intermediate levels of blur in each neuron, indicating that blur-tolerant shape identity may be decoded from a V4 population response. This is significant toward visual computation in natural environments, as defocus, due to a finite depth of field or improper accommodation, may introduce optical blur within a scene; it has been argued that the visual system responds very differently to artificial versus naturalistic stimuli presented under blur at various depths from the plane of focus[12,15]. For example, blur may aid in solving the correspondence problem of binocular disparity: a V4 neuron tuned for relative depth from the focal plane should respond strongly to a single blurred edge presented in depth and be suppressed when sharp edges of two physical objects are presented on the focal plane, which coincidentally fall within displaced binocular receptive fields. Such a mechanism would be consistent with previous work suggesting that binocular V4 neurons solve the correspondence problem by attenuating disparity signals that do

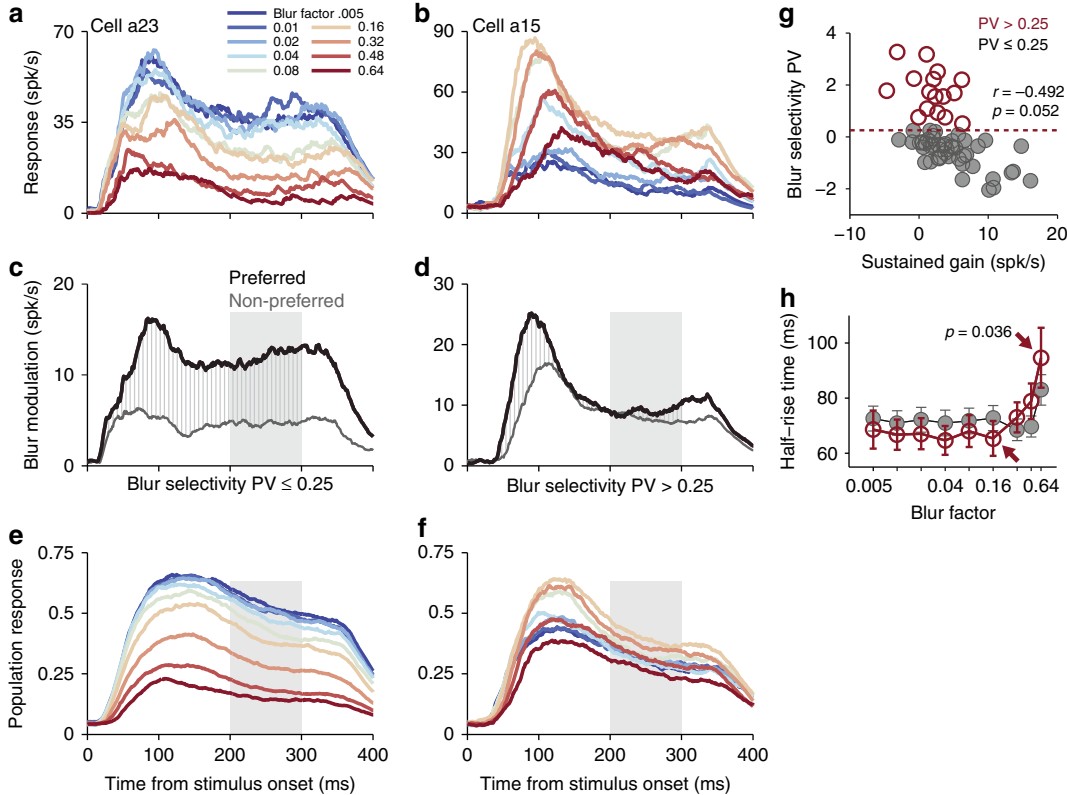

**Fig. 8** Response dynamics differ with respect to blur selectivity. **a**, **b** PSTH of preferred stimuli for low (blue) to high (red) blur factors for sharp-selective and blur-selective cells. **a** For the sharp-selective cell, increasing blur decreases neuronal response throughout the stimulus presentation interval. **b** For the blur-selective cell, blur modulation is transient. **c**, **d** Blur modulation (y-axis), calculated as the s.d. of responses with respect to blur, for preferred and non-preferred shape stimuli from example cells in **a** and **b**. The difference between the blur modulation timecourse (hatching) of preferred and non-preferred shapes captures the period in time at which blur multiplicatively scales shape-selective responses. **e**, **f** Average normalized PSTHs demonstrating distinct response dynamics between these cell groups. **g** The principle value of blur selectivity (Fig. 2e) plotted as a function of the average difference in blur modulation between preferred and non-preferred shape stimuli across the sustained period (200–300 ms; shaded region of **e**, **f**). **h** For each blur factor, mean latency to half-of-maximum response (half-rise time) for each cell group. Error bars denote s.e.m. A significant increase in response latency (t-test, p = 0.036) occurs across cells selective for intermediate blur (PV 0.25) between blur factors 0.16 and 0.64 (arrows)

not agree in SF content[29]. Thus, V4 plays a critical role in not only detecting shapes subjected to blur for judgments of object depth, but also for segmenting naturalistic scenes into blur-invariant object representations.

While we did not find blur selectivity to significantly correlate with many physiological properties of neurons within our data set, i.e., receptive field eccentricity, preference for size, color, luminance, or cortical depth of the recording site, a significant relationship was found between selectivity for intermediate blur and preferred stimulus saturation (Supplementary Fig. 1): neurons with higher blur selectivity tend to prefer shape stimuli defined by low chromatic contrasts. Given then that blur is a critical cue for the perception of shadows, this finding is consistent with blur-selective neurons acting as 'shadow detectors', since shadows cast across physical objects produce blurred edges dominated by luminance contrast[14].

Shape theorists and psychophysicists have long argued that attached and cast shadows, formed on the occluding object or another surface, respectively, contribute to the perception of 3D shape and scene understanding. For example, the relative position of shadows may be used to infer the relative location of scene illuminants[30–34]. However, even though dark and blurry boundaries may be quickly identified as shadows[35], perceptual judgments on these shadows are difficult and slow[14, 36]. While some have argued that the poor access to shadow-specific

information is due to low-level shadow detection and discounting[14], others have proposed a higher-level process related to the perceptual segregation of physical objects from nuisance factors related to illumination[36]. Our results identify V4 as a plausible locus for processing shadows within the ventral stream, where shape and blur information coalesce in the activity of individual neurons. While the early emergence of blur-selective responses in V4 could underlie the rapid detection of shadows, more experiments are needed to determine whether V4 differentially encodes shadow versus non-shadow blurred boundaries, and how this difference could underlie the limited salience of shadows during perceptual judgments. An adaptive stimulus presentation protocol to first identify shape-selective and blur-selective V4 neurons, followed by an investigation of how these neurons respond to naturalistic images, will give insight into how neurons selective for blur and/or shape participate in the perceptual organization of natural scenes.

We note, however, that the descriptive model presented here does little to explain how computations encoding shape and blur arise in vivo. One model of V4 activity, the Spectral Receptive Field (SRF) model, is a tempting candidate to explain the band-selective nature of blur tuning, but previous work has shown such linear combinations of spectral power as incapable of explaining V4 shape selectivity: SRF's are unable to disambiguate stimuli that contain identical spectral power, such as any shape and its 180°-

rotated counterpart[25]. Therefore, further modeling studies are required to determine how V4 selectivity for shape and blur could be constructed from upstream populations. Fortunately, the dynamics of blur selectivity reported here provide key insights into potential underlying mechanisms. Our results demonstrate that preference for intermediate levels of blur arise early, approximately 60 to 100 ms after stimulus onset, comparable to the time at which shape selectivity arises in V4[37]. Furthermore, this activity is transient, lasting until approximately 150 to 200 ms after stimulus onset. One candidate blur-selective circuit consists of a simple difference of SF power within V1, where intermediate spatial frequency responses are inhibited by activity selective for higher spatial frequencies. This is a markedly different computation from simple band-pass SF tuning, which would not be associated with stronger responses for intermediate blur levels, since blurring never increases SF power within a stimulus. While such a spectral difference model[38] cannot capture shape selectivity of V4 neurons[25], computations similar to SRF's may explain blur tuning, consistent with contrast energy models of blur discrimination[17]. Documented effects of high-SF gratings inhibiting V1 activity[39] (see also refs. [40–45]) may underlie these computations, though the extent of such SF-based modulation in naturalistic contexts, e.g., focal blur or illumination shading, remains unknown. Such blur signals, selective for stimuli containing strong intermediate-SF and little higher-SF content, could then bypass V2 to apply fast gain reduction in shape-selective V4 units. Recurrent inhibition, either within V4 or between V4 and previous areas, would then suppress the contribution of blur over sustained periods. Alternatively, tuning for intermediate blur could arise from latent normalization of V1 activity, including faster magnocellular inputs, as high-SF responses are removed. It is unknown, however, if such a normalization-based circuit of blur tuning would reproduce these observed dynamics. It must be stressed that these circuits only describe potential mechanisms for blur tuning: blur-selective activity must then converge upon shape-selective activity within V4 to produce separable shape and blur tuning.

Computational studies have often argued that ideal representations within the earliest stages of visual processing are general-purpose codes, supporting a diversity of tasks, from which perception can emerge[10]. Consistent with this argument, V1 receptive fields, tuned for local orientation[46–48] and spatial frequency[49, 50], form a wavelet-like representation of visual space[51]. Further, it has been shown that natural scenes can be efficiently decomposed into scale-localized and space-localized Gabor-like bases, which are selective for orientation, remarkably similar to the receptive fields of V1 neurons[52, 53]. Local populations of V1 units therefore form a complete and efficient neural representation of naturalistic scenes[54]. Beyond V1, however, sensory representations of higher visual areas are thought to instead participate in solving specific visual tasks. For example, face-selective neurons in inferotemporal (IT) cortex facilitate face recognition[55], and border-ownership signals in V2 may underlie figure–ground organization[56]. Thus, rather than a general-purpose code, representations within each module beyond V1 appear to reflect the computations required to solve well-defined problems. Previous studies of V4 have shown that may neurons explicitly encode the curvature of object boundary fragments, thought to provide a structural code for complex object shape[57]. Our demonstration of tuning for both shape and blur is especially significant since such an encoding framework may provide a sufficient representation of naturalistic scenes[10]. Therefore, in addition to supporting a neural code for object recognition, V4 may also efficiently encode visual scenes for use in higher visual areas, e.g., IT. Furthermore, while the representations of V1 and V4 may both be complete, single-unit V4 activity, unlike V1,

includes an explicit code of object-centric boundary conformation. This interpretation is consistent with V1 encoding 'stuff' and V4 building an intermediate annotation of 'things', both of which are likely prominent in higher ventral computations[58]. However, the notion of shape and blur underlying a sufficient representation of natural images does not imply these features to be the sole dimensions of selectivity within V4; while higher ventral visual areas like IT may in fact decode the entirety of an image from V4, an overcomplete representation incorporating additional visual features may further benefit complex visual tasks such as object identification or scene categorization.

## Methods

**Animals and surgery.** Two rhesus monkeys (*Macaca mulatta*, one female and one male) were surgically implanted with custom-build head posts attached to the skull with orthopedic screws. After fixation training, a recording chamber was implanted; a craniotomy (≈10 mm diameter) was subsequently performed to expose dorsal area V4. See ref. [37] for detailed surgical procedures. All animal procedures conformed to NIH guidelines and were approved by the Institutional Animal Care and Use Committee at the University of Washington.

Animals were seated in front of a CRT monitor at a distance of 57 cm and were trained to fixate on a 0.1° white dot within 0.5–0.75° of visual angle for 3–5 s for water reward. Eye position was monitored using a 1 kHz infrared eye-tracking system (Eyelink 1000; SR Research). Stimulus presentation and animal behavior were controlled by customized software PYPE (originally developed in the Gallant Laboratory, University of California, Berkeley, Berkeley, CA). Each trial began with the presentation of a fixation spot at the center of the screen. Once fixation was acquired, four to six stimuli were presented in succession, each for 300 ms, separated by interstimulus intervals of 200 ms. Stimulus onset and offset times were based on photodiode detection of synchronized pulses in the lower left corner of the monitor.

**Data collection.** During each recording session, a single transdural tungsten microelectrode was lowered into cortex with an electromechanical microdrive system (Gray Matter Research). Electrode signals were amplified and single-unit activity was isolated using online spike sorting (Plexon Systems). Electrode penetrations targeted dorsal V4 from structural MRI scans localizing the prelunate gyrus. Single-unit waveforms that responded briskly to the onset of shape stimuli were identified for further recording. After data collection, spikes were sorted offline with custom software (Plexon Systems) and exported for analysis.

**Visual stimulation.** For each recorded neuron, we first characterized the preferred RF location, size, luminance contrast and chromaticity with custom shape stimuli under mouse control. Shape stimuli were presented on an achromatic gray background of mean luminance 5.4 cd/m². Foreground luminance was chosen from four values (2.7, 5.4, 8.1, or 12.1 cd/m²) that were darker, equiluminant, or brighter than the background; chromaticity was selected from 25 gamma-corrected hues spanning the CIE color space[58]. Next, we assessed shape selectivity with a standard set of 366 shape stimuli generated by rotating 51 shapes (Fig. 1a) by increments of 45° (Fig. 1b), and discarding duplicates due to radial symmetry. The design of these stimuli is described in detail elsewhere[22]. All stimuli were presented in the center of the RF and were scaled such that all parts of the stimuli were within the estimated RF of the cell: the largest shape stimulus typically had outermost edges at a distance of 75% RF radius. Stimuli were presented in random order without replacement with three repeats per stimulus.

To assess how stimulus blur influences V4 responses, we identified 5–8 shape stimuli that evoked a range of responses, from weak (non-preferred) to strong (preferred), during the shape screen described above; we then studied responses to these shapes subjected to different levels of blur. As V4 neurons respond selectively for shape orientation[22], it was often the case that preferred and non-preferred stimuli were chosen to be 180° rotation pairs of the same shape. This had the added benefit of controlling for spectral content, since such stimuli have identical spectral power[25]. Additionally, neutral curvature (circular) stimuli were also included for most neurons. Each of the chosen shapes were presented at up to 9 blur factors along an approximately exponential scale, i.e., $\beta \in$ {0.005,0.01,0.02,0.04,0.08,0.16,0.32,0.48,0.64} (Fig. 1e). Stimuli were blurred by convolving the discretized raster image with a circular 2D Gaussian blur kernel. The kernel standard deviation, denoted by a blur factor $\beta$, is written in units relative to the radius of the large circle (Fig. 1a, black arrow). Due to the limited color gamut of the display, dithering was employed by noising each pixel with zero-mean Gaussian noise with s.d. of two 8-bit greylevels. The resulting pixel intensities were linearly interpolated between background luminance and preferred color, rounded to the nearest calibrated RGB values. Finally, to prevent aliasing, stimuli were down-sampled by a factor of two. During the shape screen to assess shape selectivity, sharp stimuli were presented under a minimal blur factor of $\beta = 0.005$. Blur stimuli were randomly chosen without replacement with a median of 20 repetitions.

**Control experiments.** On a subset of cells, we conducted control experiments to evaluate whether preferred responses to intermediate blur factors could be explained on the basis of selectivity for stimulus size or stimulus contrast. Because stimulus blur increases the number of pixels distinct from the background (Fig. 4a), preference for a specific level of blur could represent preference for stimulus size. If this were the case then blur preference will depend on the absolute size of the stimulus. To control for this, we presented blurred stimuli at multiple sizes and asked whether blur preference depended on the size of the stimulus. Size control stimuli were generated from up to three exemplar blur factors, consisting of the extremal blur levels $\beta = 0.005$ and 0.64, along with an intermediate factor, typically the blur factor that evoked the strongest responses from the neuron. Shape stimuli at each of the blur factors were resized with scaling factors of 0.9 and 1.1, i.e., scaled by $\pm 10\%$. These factors were chosen for their visual correspondence to stimuli subjected to intermediate blur factors of 0.08 and 0.16, i.e., Fig. 4a, and approximate contours generated from luminance thresholds at 1/3 and 2/3 of maximum stimulus intensity under blurring of $\beta = 0.16$. Each stimulus was presented randomly with 10 to 20 repeats. A model-based analysis of size control data was performed in two ways. First, for each neuron an APC model was fit to the full set of shape data used for shape selectivity characterization. Contours of scaled and blurred stimuli were then constructed via level set contours as described for the curvature modification analysis (see Fig. 6 for details). Fitted APC models were then used to predict responses to size control stimuli. We found that the overwhelming majority of cells predicted size control responses significantly below fit performance across shape selectivity data, suggesting that responses to scaled and blurred stimuli cannot be explained on the basis of selectivity for boundary curvature alone. In a second analysis we evaluated the ability of size or blur information to explain the variance of responses to control data. Here, for each neuron we fit APC×B and APC×S models to responses of scaled and blurred stimuli, where S is a Gaussian tuning function of size computed from the arc length of a stimuli's level-set contour. Again, responses were overwhelmingly better fit across our population by the joint shape and blur-selective APC×B model, despite the APC×S model having identical number of free parameters and functional form.

To evaluate whether preference for an intermediate level of blur could be explained on the basis of preference for average stimulus contrast, we also studied responses to contrast control stimuli generated by first computing mean stimulus intensity across the interior of a blurred shape stimulus, where shape interior is defined by the half-contrast level set. Then, for each blur factor, a control stimulus was generated with a foreground intensity equal to the mean intensity within this level set (Fig. 5). Control stimuli were subjected to only the minimal blur factor of $\beta = 0.005$, resulting in stimuli with sharp boundaries of reduced figure–ground contrast.

**Analysis and model fitting.** Neural responses to individual stimuli were calculated as the mean firing rate observed during stimulus presentation, 300 ms in duration with a 50 ms lag relative to onset, averaged across repeats. Peristimulus time histograms were computed for each stimulus by filtering spike rasters with a centered (noncausal) decaying exponential filter consistent with a membrane's integration time constant (37 ms).

Preferred-shape blur tuning curves (see Figs 3 and 5) were constructed by first identifying preferred shapes, i.e., sharp stimuli ($\beta = 0.005$) that evoked a response greater than mean across shapes (2–4 preferred shapes per cell), and averaging across preferred shapes for each blur factor. A cubic spline is then fit to average preferred shape responses. Center-of-mass (CoM) was calculated by integrating preferred tuning curves across blur factors in log space (or similarly in intensity-matched factors during intensity control analysis), and returning the median cumulative factor. Significance of CoM measurements from intensity-matched controls was determined by bootstrapping estimates of tuning curves sampled from response distributions of recorded means and variances from each stimulus (100 repetitions) under a two-way $t$-test of unequal variance.

The APC model has been shown to accurately capture shape tuning properties of single-unit V4 spike-rate responses[22, 25]. Here, each shape stimulus is represented by 4–8 points in the space of angular position $\theta$ and curvature $\kappa$, corresponding to locations of curvature inflection along the contour. In particular, a stimulus $\Gamma = (\gamma^1, \gamma^2, \dots, \gamma^n)$ is represented by $n$ critical points $\gamma^i = (\gamma^i_\theta, \gamma^i_\kappa)$. The model predicts responses to shape stimuli by evaluating a Gaussian energy function (von Mises in periodic angular position) at each of these points, and returning the maximum. When applicable, $\Gamma$ is augmented with a blur factor $\gamma_\beta$ proportional to the kernel size of a Gaussian-blurred shape stimuli. Thus,

$$\text{APC}(\Gamma; \omega, \alpha, \mu_\theta, \sigma_\theta, \mu_\kappa, \sigma_\kappa) = \omega + \alpha \exp\left(\frac{\cos(\gamma^i_\theta - \mu_\theta)}{\sigma_\theta} - \frac{(\gamma^i_\kappa - \mu_\kappa)^2}{\sigma^2_\kappa}\right) \quad (1)$$

describes a model over dimensions of angular position and curvature. Note that the tuning peak $\mu$ and width $\sigma$ are represented along each selectivity dimension $\theta$ and $\kappa$. Further, baseline parameter $\omega$ captures spontaneous activity in the absence of stimulation, and gain $\alpha$ is fit to produce maximal responses for preferred stimuli. We extend the APC model to also predict neural responses as a function of boundary blur. Here, blur selectivity is modeled as Gaussian in the logarithm of

blur factors, i.e.,

$$B(\Gamma; \mu_\beta, \sigma_\beta) = \exp\left(-\frac{(\log(\gamma_\beta) - \mu_\beta)^2}{\sigma^2_\beta}\right). \quad (2)$$

Again note preferred blur factor $\mu_\beta$ and blur tuning width $\sigma_\beta$. Every model is fit to minimize squared error between mean firing rate $\vec{r}$ evoked by stimuli $\vec{\Gamma}$, averaged across repetitions, and the responses predicted by the model. For example, a fit $\Theta^*$ of neural data to an angular position, curvature, and blur (APCB) model is written

$$\Theta^* = \arg\max_\Theta \left\| \max_i \left(\text{APC}(\vec{\Gamma}; \Theta) * B(\vec{\Gamma}; \Theta)\right) - \vec{r} \right\|_2, \quad (3)$$

such that the model predicts a response to any stimulus as the maximum of each critical point $\gamma^i$ evaluated under that model. The optimal model therefore minimizes the $L_2$ norm between recorded $\vec{r}$ and predicted responses of stimulus set $\vec{\Gamma}$. Model fitting is complicated by the fact that optimization is highly non-convex. While standard gradient descent methods are quick to converge, solutions are typically only locally optimal: we employ a repeated randomized initialization procedure to approximate globally optimal fits, described elsewhere[25].

Model selection is performed using leave-one-out cross-validation. APC and APCB models were trained on all but one blurred stimulus and then used to predict the hold-out response. This procedure was repeated for all blurred stimuli for each cell. Training and testing error was computed by averaging NRMSE error for every training session (i.e., every stimulus in the blur data set independently for each neuron). To measure significance of validation performance for each cell we estimate the distribution of average hold-out prediction error for the APCB model relative to the APC model. Significance is determined via paired $t$-test of relative testing error, paired across blurred stimuli for each neuron.

For each blurred stimulus, we computed the boundary contours by binary-thresholding images at a range of intensity levels. These contours were then represented as points in the 2D space of curvature and angular position. Each shape and its blurred counterparts were represented by the same set of angular position values, but the curvature values were reduced under blur (Fig. 6a, b).

To compare model fit performance we measure prediction accuracy to a subset of shape stimuli. For each stimulus, the set of critical points along the contour was evaluated under a trained APC model to produce a response prediction $\vec{p}$. We compute the normalized root-mean-squared error (NRMSE) between the predicted and recorded responses $\vec{r}$, i.e.,

$$\frac{\| \vec{p} - \vec{r} \|_2}{\max(\vec{r}) - \min(\vec{r})}. \quad (4)$$

NRMSE training estimates were computed via bootstrapping: a random subset of shape stimuli were selected, equal in number to the number of blurred stimuli recorded, and the NRMSE is again calculated between the best-fitting APC model prediction to responses elicited from sharp stimuli from the entire shape set.

**Data availability.** The data and analysis code that support the findings of this study are available from the corresponding author upon reasonable request.

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

## Acknowledgements

We thank W. Bair and J. Elder for helpful discussions and comments on the manuscript and A. Fyall for assistance with animal training and data collection. Technical support was provided by the Bioengineering group at the Washington National Primate Research Center. This work was funded by NEI grant R01EY018839 to A.P., Vision Core grant P30EY01730 to the University of Washington, P51 grant OD010425 to the Washington National Primate Research Center, Natural Sciences and Research Counsel of Canada PGS-D to T.D.O., and University of Washington Computational Neuroscience Training Grant to T.D.O.

## Author contributions

T.D.O. and A.P. contributed experiment design, analysis methodology, and manuscript preparation; T.D.O. and A.N. contributed data collection. T.D.O. contributed data analysis and modeling. All authors contributed to interpretation of findings and manuscript revisions.

## Additional information

**Competing interests:** The authors declare no competing financial interests.

