## [Peer Review File · Nature Communications]

Reviewers' comments:

Reviewer #1 (Remarks to the Author):

This paper attempts to define a real-life parameter describing visual statistics, namely blur, as being relevant and important to understanding the encoding seen at a cellular level in area V4. While it is certainly clear that no previously investigators have explicitly measure blur tuning, the critical question with regard to the significance of this paper is whether a study of blur tuning yields new insights into V4 receptive fields. Unfortunately, given the extensive literature of spatial frequency tuning both in V4 and in earlier areas, the authors do not convincingly address this question. There are multiple challenges: blur, shape and contrast are not separable, and some of the controls the authors provide in these respects are challenging to understand and could potentially be addressed by introducing or refining non-linearities in a well-established model of V4 like the spectral receptive fields defined by the Gallant lab. Unfortunately, SRFs which were shown to explain both responses to natural stimuli and shape/curvature sets such as those employed in this study, are largely ignored by this paper. We therefore do not know whether a previous model, based on psychophysically and physiologically data pointing to spatial frequency tuning, cannot completely explain the authors' results in an elegant and simple way.

1. This paper is fundamentally based on the premise that blur is ecologically relevant parameter that has not been sufficiently characterized by previous receptive field analyses. Because mathematically the blur operation is equivalent to a SF filter (for example, convolution with a Gaussian spatial filter is a Gaussian filter in frequency space), the only way that a classic spatial frequency approach (such as the spectral receptive field approach from the Gallant lab, ref 35) cannot apply is if there are interesting non-linearities. But the SRF already has non-linearities in it, and has been show to explain both responses in the curvature-shape stimulus space and natural stimuli, and in this paper, an explicit RF predictive modeling incorporating non-linearities is never attempted. Spatial frequency channels have a long history both psychophysically and physiologically, and yet this paper seems almost ignore this history by suggesting that, for example, band-pass SF tuning cannot possibly explain preferred blur levels that are intermediate. As near I can tell, the evidence for this seems to rest on anecdotal examples like Fig 3c, in which the high blur factor responses to some stimuli are different than the high-blur responses to other stimuli. I don't find this very convincing or systematic, especially since at very high blur factors (as the authors verify) responses have to be similar because you've eliminated shape.

2. We know that classically defined band-pass SF tuning would create intermediate blur tuning, so then the argument has to be that the blur tuning cannot be fully explained by band-pass SF. The onus is therefore on the authors to demonstrate this and show that an SRF cannot explain these results. With this data, one could potentially refine V4 receptive field models, but without it, it is not even clear that a "blur"-based approach provides significant new insight.

3. I found the argument of curvature attenuation as a contributor to blur selectivity difficult to follow. I'm also not certain how this could work given the lack of independence between shape and blur, since shape selectivity can only exist with low levels of blur.

4. One argument made regarding the significance of blur tuning is that taking it into account seems to improve model fits (Figure 7). But, given the increase in the number of free parameters, doesn't this have to be true? In this respect, it seems a much more appropriate analysis would AIC, BIC (or even chi-square if the models are nested) which takes into account the number of free parameters.

Reviewer #2 (Remarks to the Author):

This is a study of the possible blur coding in monkey V4. It has been shown before that blurred edge or shape strongly affects our perception of a visual scene. This study is the first effort in looking for the physiological basis for such effects, thus can be of substantial interest in the field. Area V4 is also a good candidate to look at. The single-unit results are clear, and the analysis procedures are also appropriate. However, a couple of important issues, mainly experimental design and result interpretations, must be clarified before this paper can be accepted for publication.

Major:

Some basic information is missing. For example, the RF size, eccentricity and color preference of the neurons collected, and whether the "blur neurons" differ in these aspects with other V4 neurons. Whether there were any particular features in recording site, depth and clustering for these "blur neurons"? Another important RF feature is binocular disparity, which is related the question below.

As the authors note, blur is often associate with depth perception. The majority of V4 neurons also tuned to binocular disparity, mostly relative disparity. Thus, it would be very interesting to know that how a near/far tuned disparity neuron responds to a blurred shape (versus a sharp one)? It would also be useful to discuss how blur information contribute to depth/3D perception. For example, one sees strong depth order in an image containing overlaid sharp and blurred figures.

Another important question is about the stimulus set. It is difficult to rule out the possibility that the "blur-preferred" neurons could be best stimulated by a sharp shape that was not covered by the "shape set" used in this study. A blurred shape might gain extra local energy that stimulated/suppressed a subdomain of the neuron's RF. The authors assumed that the stimulus set (51 shape x 8 orientation) represented a full coverage of the shape space. But the differences between neighboring shapes are still large (Figure 1c), and the orientation steps (45 degree) were also large (Figure 1d). It might be difficult to increase the number of stimulus in the "shape set" due to the limitation of recording time. The authors may consider to adjust local blurriness of the "optimal" blurred shape to show that this shape is indeed THE best stimulus for that neuron. Such test can also reveal any particular relationships, if exist, between the blurriness and local curvature or the major axis of the shape.

It may also too ambiguous to state that this is a "joint coding" of shape and blur. Only a few neurons exhibited blur effect. The majority neurons decreased their responses simply due to loss of shape features, instead of being modulated by blur.

Minor:

Page 1 line 15: "complex complex"

Page 3, line 74-81: How the total 65 neurons were broken-down into the following categories (42, 11, 5, 10)?

Page 12 line 301: "may"

Reviewer #3 (Remarks to the Author):

Overall view

This is a novel, interesting and important study of one visual area (V4) in the neocortex of the macaque monkey. Much previous work on V4 has shown that this area is involved in 'mid-level vision', serving to derive information about 2-D shape and curvature based on input from earlier levels of visual processing. This paper extends the methods used in the authors' earlier work to test whether V4 also encodes information about image blur. Blur is a key aspect of images, but there have been few, if any, studies of blur coding in the primate brain - a truly neglected, but nonetheless very important topic. The paper is therefore novel - being probably the first study of blur coding in the brain, and it offers an important finding: The paper shows pretty convincingly that V4 neurons jointly carry information about shape and blur. In essence, some neurons respond best to sharp images, others to intermediate levels of blur, while at the same time responding differently to different contour shapes. The work does not reveal precisely how this sensitivity to blur comes about, but the Discussion offers some interesting ideas on this, and on the role of V4 in vision more broadly.

Several key main experiments are supported by additional control experiments to show that the response to blur is not really a response to other, blur-induced changes in the image structure (such as change in curvature or reduction in contrast).

Thus I'm very positive about the paper. I do however, have quite a few suggestions for improving clarity here & there, and for a few improvements in terminology.

Comments & Suggestions for clarification

1. I think the paper needs to be a bit more careful in its use of terminology concerning several key concepts: blur, boundary and contrast. On line 18, for example we read that blurred edges "exhibit a spatial gradient of contrast orthogonal to the boundary contour". This is not quite accurate though, because such edges have a spatial gradient of luminance (not contrast) across the boundary contour. Spatial luminance contrast (a normalized measure of luminance difference in an image or across a contour) is a consequence of such luminance gradients. A gradient of contrast can indeed arise at texture boundaries for example, but these are quite different and have often been called '2nd order edges'. Similar comments apply to Line 39 where we read that blur is "the gradient of boundary contrast". Actually, both increasing blur, and decreasing contrast, will decrease the luminance gradient at an edge, and so the luminance gradient alone is not a good measure of blur (because the gradient also varies with contrast when blur is held constant).

2. Further, on line 45-46, we have "Blurring a stimulus boundary, as implemented here ... broadens the gradient of contrast across the shape's edges and thus decreases effective boundary contrast." Two comments: (i) as before, it's really the gradient of luminance that we're concerned with, and (ii) it's unclear why broadening the luminance gradient should be thought to decrease contrast. Yes, the local gradients are shallower, but if contrast is determined by overall luminance difference across the edge, this does not necessarily decrease when an isolated single edge is blurred; the full contrast may exist at a coarser spatial scale, if neighbouring edges are sufficiently far away. (It's true that when multiple edges are present, then as blur increases the edges start to merge or interact and contrast does go down, but the authors don't seem to have that in mind at this point.) [It's also true that a global measure of contrast, such as RMS contrast, is reduced by blur, but that also seems less relevant here since the authors are here talking about 'boundary contrast'.]

My point is simply that terminology matters, and to help the reader we should try to get it right.

3. Line 26: "neural mechanisms that underlie the computation and representation of blur remain unclear." Yes, that's true with respect to direct neurophysiological evidence, but computational models for perception and discrimination of blur, inspired by V1 physiology, are quite well developed, and might be worth citing (eg Watt & Morgan, Vision Res, 1985; Georgeson et al, Journal of Vision 2007) in addition to the work of Watson & Ahumada (2011) already cited.

4. Lines 107-113 are intended to motivate a control experiment, but I thought the rationale here was not expressed clearly. For example, it's unclear what being different 'with respect to an 8-bit stimulus gamut' might mean. And it begins with (what seems to me) a rather unlikely hypothesis about what defines a luminance boundary: "that boundaries are identified from the contiguous region of a certain stimulus intensity". [It seems unlikely for many reasons, but to take just one: adding a modest amount of 2-D masking noise to one of these shapes would have little effect on boundary perception, but would completely eliminate the existence of any such 'contiguous region of a certain stimulus intensity', because every pixel would then be different from every other.]. Perhaps I've misunderstood something, but any clarification would be welcome. Lines 112-113 are clearer than what precedes them.

Re line 115, Fig 5 rather adds to the confusion because panel a implies that the shapes were white on a black background, while panel b shows the reverse (dark figures, white background). The Method section later reveals that both negative & positive contrasts were used, but it would be useful to clarify this with a few words in the main text or figure 5 legend. On the other hand, description of the results (lines 114-129) is clear and easily grasped.

5. Line 171-172: "We then augment the APC model to include a blur-selective term, taken to be log-normal in blur factor beta, that scales shape-selective responses." This is a crucial statement of the APCB model favoured by the authors & implied by the data. It's fine as far as it goes, but it might help the reader for this statement to be reinforced by the sort of simple algebraic relation given on Line 184 for the not-favoured APC+B model. Presumably the APCB model states that cell response R to a given shape stimulus is the product of two terms, $R = APC(\text{shape}) * B(\text{blur})$, where $APC(\text{shape})$ is a function of stimulus shape but not blur, while $B(\text{blur})$ is a function of the stimulus blur level beta, but independent of shape? If I've grasped this correctly, I think it would help the reader greatly to insert such a statement on line 172. It's quite hard to deduce this from the technical methods section, so if I've got it wrong, it would still be good to insert some correct version into the main text.

6. Fig 8 and its associated text are good and important, but also hard work - perhaps necessarily so, because we are dealing with responses that exhibit a 4-way interaction between time, blur level, stimulus shape, and class of cell (blur-selective or not), with 3 output measures (spikes/s, blur modulation, and response latency). Multivariate, high-order interactions pretty much tax the human mind beyond its limits, so anything that could be added to further clarify or simplify, at least in part, would be welcome. (I know this is quite hard to do, without distorting the truth...)

7. Figs 8e,f: x-axis is labelled as "Stimulus onset time (ms)", but I think it really means "Time after stimulus onset (ms)", and would be clearer if changed.

8. Lines 234-240 are a supremely clear & succinct summary of the paper

9. Lines 255-6: "defocus, due to a finite depth of field or improper accommodation, may introduce optical (Gaussian) blur within a scene;" Delete Gaussian? I don't think that optical blur of this kind is actually Gaussian (though one might approximate it experimentally with some level Gaussian blur, as implied in line 257).

10. Lines 282-3: "One candidate blur-selective circuit consists of a simple difference of SF power within V1...". Difference of power between what and what ? I think the reader needs to know this, in order to understand what 'This blur signal' is a couple of lines later.

11. Discussion section is generally very relevant, interesting, and thought-provoking.

We thank the reviewers for their critical reading of our manuscript and thoughtful comments. Guided by their feedback, we now submit a revised manuscript that includes new analyses, major text edits (highlighted in green), one supplementary figure and one figure for review purposes. Below, we include the reviewers' specific comments (italics), interleaved with our responses. We hope that the reviewers and editors now find the manuscript suitable for publication in *Nature Communications*.

Reviewer #1

This paper attempts to define a real-life parameter describing visual statistics, namely blur, as being relevant and important to understanding the encoding seen at a cellular level in area V4. While it is certainly clear that no previously investigators have explicitly measure blur tuning, the critical question with regard to the significance of this paper is whether a study of blur tuning yields new insights into V4 receptive fields. Unfortunately, given the extensive literature of spatial frequency tuning both in V4 and in earlier areas, the authors do not convincingly address this question. There are multiple challenges: blur, shape and contrast are not separable, and some of the controls the authors provide in these respects are challenging to understand and could potentially be addressed by introducing or refining nonlinearities in a well-established model of V4 like the spectral receptive fields defined by the Gallant lab. Unfortunately, SRFs which were shown to explain both responses to natural stimuli and shape/curvature sets such as those employed in this study, are largely ignored by this paper. We therefore do not know whether a previous model, based on psychophysically and physiologically data pointing to spatial frequency tuning, cannot completely explain the authors' results in an elegant and simple way.

We thank the reviewer for raising the concern that our findings could potentially be explained by existing models of spatial frequency tuning in V4. We have previously demonstrated that SRF models *cannot* explain tuning for boundary curvature (Oleskiw et al., 2014); because of this, in the first version of our manuscript we did not consider SRF models as a candidate for explaining joint tuning for shape and blur. In our revised Discussion (see lines 303-307 and 313-317) we now elaborate why SRF models cannot explain tuning for boundary curvature and therefore cannot explain joint tuning for shape and blur.

The spectral receptive field model (David et al, 2006) does not explain tuning for boundary curvature (Oleskiw et al., 2014). As the reviewer aptly notes, Gallant and colleagues did speculate that SRF models may explain tuning for boundary curvature (David *et al.* (2006, [38])). Upon further analysis, however, we have shown this hypothesis to be incorrect (Oleskiw *et al.* (2014, [25])). Adapted from our own study, Figure R1 (see below) readily demonstrates that SRF models will predict identical responses to stimuli that have the same power spectrum while differing in phase. In contrast, curvature-tuned V4 neurons overwhelmingly exhibit significantly different responses to 180° rotated versions of a shape, which exhibit precisely identical spectral power. For a demonstration of this in our present work consider cell a15 of Figure 2d. In this plot the preferred shape (shape #1, top dark-red) and a non-preferred shape (shape #6, blue) are 180° rotated versions of each other. Due to the phase inversion property of Fourier transforms, it can be shown analytically that both of these stimuli have an identical spectral power regardless of blur magnitude (see Oleskiw *et al.* 2014 for a derivation of this). Inconsistent with SRFs, these stimuli evoked significantly different firing rates across low and intermediate blur levels. Thus, the activity of this neuron cannot be explained by *any* SRF model, even if additional nonlinearities were included.

We refer the reviewer to Oleskiw *et al.* (2014), where we pursue this point further and conduct detailed modelling of V4 data to demonstrate why SRF models cannot explain curvature tuning. Further, in personal communication Jack Gallant has conceded that the original speculation of their 2006 paper was incorrect.

Figure R1: **Spectral receptive fields do not account for shape selectivity**, adapted from Oleskiw *et al.* (2014). (a) Three example shape stimuli, their spectral power transformation, and sub-sampled spectral power representation used for SRF modelling. Note the bottom two stimuli, being 180° rotation pairs, have precisely identical power spectra and thus are indistinguishable to all SRF models. (b) SRF models predict very similar responses (correlation near 1, diamonds in Figure R1) to spectrally-identical stimuli (up to Poisson noise), much higher than what is observed in a population of V4 neurons (dots, Figure R1) or what is predicted by APC models of shape selectivity (open circles, Figure R1).

Ultimately, we agree with the Reviewer: the significance of our paper lies in the fact that our results cannot be readily explained by existing models of spatial frequency tuning in V4.

This paper is fundamentally based on the premise that blur is ecologically relevant parameter that has not been sufficiently characterized by previous receptive field analyses. Because mathematically the blur operation is equivalent to a SF filter (for example, convolution with a Gaussian spatial filter is a Gaussian filter in frequency space), the only way that a classic spatial frequency approach (such as the spectral receptive field approach from the Gallant lab, ref 35) cannot apply is if there are interesting nonlinearities. But the SRF already has nonlinearities in it, and has been show to explain both responses in the curvature-shape stimulus space and natural stimuli, and in this paper, an explicit RF predictive modeling incorporating nonlinearities is never attempted. Spatial frequency channels have a long history both psychophysically and physiologically, and yet this paper seems almost ignore this history by suggesting that, for example, band-pass SF tuning cannot possibly explain preferred blur levels that are intermediate. As near I can tell, the evidence for this seems to rest on anecdotal examples like Fig 3c, in which the high blur factor responses to some stimuli are different than the high-blur responses to other stimuli. I don't find this very convincing or systematic, especially since at very high blur factors (as the authors verify) responses have to be similar because you've eliminated shape.

In the revised Discussion (see 257-259, 313-317) we now consider in detail whether band-pass SF tuning can explain tuning for blur. Here, we argue that simple band-pass SF tuning *cannot* explain tuning for intermediate levels of blur because a non-blurred (sharp) stimulus retains intermediate SF power yet fails to drive neurons tuned for intermediate blur factors. In particular, a neuron with band-pass SF tuning would be expected to respond similarly to both sharp stimuli and intermediately-blurred stimuli, since each have identical energy at relevant SF bands: blurring does not *increase* spectral energy in *any* band. Tuning for intermediate blur could arise, however, if band-pass SF tuned responses are normalized by the total spectral energy across all bands (see lines 318-324). Another simple blur-selective circuit consists of a simple difference of SF power within V1, where intermediate spatial frequency responses are inhibited by activity selective for higher spatial frequencies.

Thus, while SRF models (David et al., 2006) cannot capture curvature tuning of V4 neurons (Oleskiw et al., 2014), computations similar to SRF's could underlie tuning for blur which could then modulate the gain of shape-selective V4 units to produce joint tuning for shape and blur observed here. We now discuss this novel computation, not described previously in the literature, as a possible mechanism that underlies our results (see lines 303-318).

We know that classically defined band-pass SF tuning would create intermediate blur tuning, so then the argument has to be that the blur tuning cannot be fully explained by band-pass SF. The onus is therefore on the authors to demonstrate this and show that an SRF cannot explain these results. With this data, one could potentially refine V4 receptive field models, but without it, it is not even clear that a "blur"-based approach provides significant new insight.

As articulated in the responses above, and in the revised discussion (see lines 318-324), band-pass spatial frequency tuning combined with normalization is likely a necessary feature of blur selectivity. However, band-pass SF tuning is not sufficient to explain the critical and novel finding of our work: a combination of blur- and shape-selectivity observed within single V4 neurons. Our demonstration of shape-specific blur modulation implies that intermediate spatial frequency tuning is an insufficient explanation for our findings, since boundary conformation cannot be decoded from oriented spectral power alone (see Oleskiw et al., 2014 and our explanations above).

I found the argument of curvature attenuation as a contributor to blur selectivity difficult to follow. I'm also not certain how this could work given the lack of independence between shape and blur, since shape selectivity can only exist with low levels of blur.

We have revised the relevant sections to more clearly articulate why curvature is modified under blur, and thus better motivate the need to preform our curvature control (see lines 138-144). The lack of independence between boundary curvature and blur was precisely our motivation for performing the curvature attenuation control analysis. Specifically, when stimuli are blurred, boundary curvature can be modified, and therefore the preference for intermediate blur could simply be due to curvatures sampled by blurred stimuli that were not sampled by their sharp counterparts. In our experiments we studied shape responses over a large range of blur values, finding shape selectivity to occur not just for low blur but also for intermediate blur values across the range tested (see Fig. 2 for examples of this). Thus, even though shape selectivity is largely absent at high levels of blur, we can examine this question effectively across low and intermediate blur factors.

One argument made regarding the significance of blur tuning is that taking it into account seems to improve model fits (Figure 7). But, given the increase in the number of free parameters, doesn't this have to be true? In this respect, it seems a much more appropriate analysis would AIC, BIC (or even chi-square if the models are nested) which takes into account the number of free parameters.

We thank the reviewer for this thought-provoking comment. Because AIC and BIC require model linearity assumptions that are not applicable in our case, we now assess model significance using a method of cross-validation. Our motivation here, as shared by the reviewer, is to examine to what extent each model can generalize across observed data to explain variance without overfitting. Thus we perform a leave-one-out cross-validation of both APC and APCB models for all cells, results of which have been included in Figure 7b,d,g. We quantify performance on training and testing datasets by computing the difference in average NRMSE error of the APC model relative to the APCB model (*i.e.* APC - APCB error, Fig. 7b). Not surprisingly, since the APCB model is a generalized version of the APC model having more parameters, the APCB model has a positive relative training error for all cells (Fig. 7b, *x*-axis). Importantly, all but one of our cells have a positive relative testing error (Fig. 7b, *y*-axis), implying that the APCB model is overwhelmingly better equipped to capture responses of neurons to blurred stimuli and this is not just a result of overfitting with additional parameters.

Next, to examine the *significance* of this reduction in prediction error, for every neuron in our population we perform a paired t-test across hold-out stimuli. Specifically, for each cell we estimate the distribution of prediction improvements of the APCB model, relative to the APC model, paired across blurred stimuli. In Figure 7g we plot the *p* value of each cell as a function its blur selectivity PV; across our population 12 neurons fail to exhibit a significant difference in prediction performance ($p > .05$). Importantly, neurons selective for intermediate blur (large positive blur PV) demonstrate a significant improvement in prediction error (Fig. 7d), consistent with blur-selective responses better explained by the APCB model.

Thus, while the APCB model contains additional parameters, and is indeed a generalization of the APC model, the inclusion of a multiplicative blur tuning function significantly increases our ability to explain the variance of responses to blurred stimuli in the majority of neurons in our population. We thank the reviewer for noting the weakness in our original model selection procedure, and find this new validation-based method of model selection to be a superior alternative to others discussed above. The associated results, methods, and Figure 7 caption text have been modified to describe our approach and findings.

Reviewer #2

This is a study of the possible blur coding in monkey V4. It has been shown before that blurred edge or shape strongly affects our perception of a visual scene. This study is the first effort in looking for the physiological basis for such effects, thus can be of substantial interest in the field. Area V4 is also a good candidate to look at. The single-unit results are clear, and the analysis procedures are also appropriate. However, a couple of important issues, mainly experimental design and result interpretations, must be clarified before this paper can be accepted for publication.

Some basic information is missing. For example, the RF size, eccentricity and color preference of the neurons collected, and whether the "blur neurons" differ in these aspects with other V4 neurons. Whether there were any particular features in recording site, depth and clustering for these "blur neurons"? Another important RF feature is binocular disparity, which is related the question below.

We thank the reviewer for this comment and now include supplemental Figure S1 to illustrate the relationship between blur tuning of recorded neurons and RF eccentricity, stimulus size, stimulus color, luminance contrast, and cortical depth of recorded neurons. The only stimulus factor associated with a statistically significant correlation with blur selectivity was stimulus saturation: in Figure S1g we see neurons that preferred low saturation chromatic stimuli were tuned to intermediate blur stimuli. This result is consistent with blur-tuned neurons acting as 'shadow detectors'

and we now mention this in the discussion (see lines 279-285). Within our limited dataset, however, it is difficult for us to make any concrete claims about the clustering of blur selective neurons.

As the authors note, blur is often associate with depth perception. The majority of V4 neurons also tuned to binocular disparity, mostly relative disparity. Thus, it would be very interesting to know that how a near/far tuned disparity neuron responds to a blurred shape (versus a sharp one)? It would also be useful to discuss how blur information contribute to depth/3D perception. For example, one sees strong depth order in an image containing overlaid sharp and blurred figures.

As stated by the reviewer there are many fascinating questions surrounding blur in cuing percepts of depth. There is much existing research on this, as a number of psychophysical studies have investigated how defocus blur can be used for depth estimation. We have revised our introduction and discussion to emphasize this point, including associated references that explain in detail the relationship between defocus blur and depth perception (see lines 27-30 and 267-276).

Our lab does not have the necessary equipment to present dichoptic stimuli so we cannot study how disparity-tuned neurons would respond to blurred stimuli in depth. However, as noted in our revised Discussion, we can speculate that disparity-tuned neurons may use blur information to help solve the correspondence problem: a disparity tuned neuron ought to respond more strongly to blurred dichoptic stimuli than sharp stimuli, since blurred stimuli would correspond to a single physical object presented in depth from the focal plane, whereas sharp stimuli would merely indicate edges of two different physical objects at the focal plane which coincidentally fall on binocular receptive fields displaced in each eye. This conjecture is consistent with the literature, as Kumano *et al.* (2008, [29]) suggested binocular V4 neurons integrate spatial frequency information to solve the correspondence problem. Since our experiments presented stimuli on the focal plane, we likely under-sampled blur-selective neurons which were also tuned for disparity. We are eager to collaborate with a lab equipped to preform such experiments to quantitatively explore blur and disparity computations.

Another important question is about the stimulus set. It is difficult to rule out the possibility that the "blur-preferred" neurons could be best stimulated by a sharp shape that was not covered by the "shape set" used in this study. A blurred shape might gain extra local energy that stimulated/suppressed a subdomain of the neuron's RF. The authors assumed that the stimulus set (51 shape x 8 orientation) represented a full coverage of the shape space. But the differences between neighboring shapes are still large (Figure 1c), and the orientation steps (45 degree) were also large (Figure 1d). It might be difficult to increase the number of stimulus in the "shape set" due to the limitation of recording time. The authors may consider to adjust local blurriness of the "optimal" blurred shape to show that this shape is indeed THE best stimulus for that neuron. Such test can also reveal any particular relationships, if exist, between the blurriness and local curvature or the major axis of the shape.

The reviewer's concern here has two components: *i*) whether V4 neurons may be sensitive to stimulus features sampled by the blurry stimulus but not the sharp shapes in our limited set, and *ii*) whether tuning for intermediate blur could result from the stimulation of an RF region not stimulated by the sharp stimulus.

To consider the possibility that neurons may be selective for curvature values present in blurred stimuli and not sharp stimuli we conducted the curvature level set analysis. Here, we explored the possibility that tuning for intermediate blur was the result of selectivity for modified curvature features that arose when sharp stimuli are progressively blurred. Our results suggest that responses to blurred stimuli are best explained by joint tuning for both boundary curvature and blur, as opposed to a model tuned for modified curvatures present in blurred stimuli. Thus, neurons selective for blur respond best to blurred stimuli rather than sharp stimuli with modified curvatures. (see Results: Controlling for attenuation of curvature and Fig. 6).

With regard to concerns of differential RF sub-region stimulation, for every neuron we assessed tuning for blur with several shape stimuli that stimulated various sub-regions of the RF. The tuning for blur was consistent across all tested shapes (see Fig. 2) suggesting that differential stimulation cannot explain tuning for blur. Furthermore, for a subset of neurons ($n = 26$ of 65) we asked whether blur tuning depended on the size of the stimulus. Specifically, by reducing and expanding the size of shape contours prior to blur, we alter the location of stimulus features on the retina. Our analysis reveals size to not explain a significant amount of response variance, consistent with features of shape and blur selectivity being yoked to an object-centric reference frame. We now include these points in our revised Discussion (see lines 247-250).

The dynamic optimization of stimuli, suggested by the reviewer, is a strategy used in many past studies (e.g. Kobatake and Tanaka, 1994; Yamane et al., 2008) to identify ‘optimal’ stimuli for a given neuron. To be successful, however, such experiments will need hundreds of trials beyond the already large stimulus sets that we currently employ. We highlight this in our Discussion as a good approach for future investigations to understand how blur selective neurons might contribute to natural vision (see 298-301). We hope that the analyses and data presented above mitigate the concern raised by the reviewer.

It may also too ambiguous to state that this is a "joint coding" of shape and blur. Only a few neurons exhibited blur effect. The majority neurons decreased their responses simply due to loss of shape features, instead of being modulated by blur.

In this study we have found a population of neurons in V4 that simultaneously represent shape and blur information, and our chosen title is a concise description of these findings. While only a subset of neurons in our V4 population are tuned for intermediate blur ($n=11$ of 65, 17%), this does not detract from the novelty or import of the finding as these neurons, together with neurons tuned for low and high blur magnitudes, form the basis of a population-level code. Furthermore, the proportion of neurons tuned to intermediate blur is comparable to the proportion of V1 cells tuned for direction of motion, which is well known to be a prominent coding dimension at the population level of primary visual cortex.

Page 1 line 15: "complex complex"

Page 3, line 74-81: How the total 65 neurons were broken-down into the following categories (42, 11, 5, 10)?

Page 12 line 301: "may"

We thank the reviewer for noting these typos. We accept these corrections and have also included the fraction of blur tuning categories observed in our population. Please see lines 76-88 for the revised description of cell categories.

Reviewer # 3

This is a novel, interesting and important study of one visual area (V4) in the neocortex of the macaque monkey. Much previous work on V4 has shown that this area is involved in 'mid-level vision', serving to derive information about 2-D shape and curvature based on input from earlier levels of visual processing. This paper extends the methods used in the authors' earlier work to test whether V4 also encodes information about image blur. Blur is a key aspect of images, but there have been few, if any, studies of blur coding in the primate brain - a truly neglected, but nonetheless very important topic. The paper is therefore novel - being probably the first study of blur coding in the brain, and it offers an important finding: The paper shows pretty convincingly that V4 neurons jointly carry information about shape and blur. In essence, some neurons respond best to sharp

images, others to intermediate levels of blur, while at the same time responding differently to different contour shapes. The work does not reveal precisely how this sensitivity to blur comes about, but the Discussion offers some interesting ideas on this, and on the role of V4 in vision more broadly. Several key main experiments are supported by additional control experiments to show that the response to blur is not really a response to other, blur-induced changes in the image structure (such as change in curvature or reduction in contrast). Thus I'm very positive about the paper. I do however, have quite a few suggestions for improving clarity here & there, and for a few improvements in terminology.

We thank the reviewer for the careful and thorough review of our work. We agree with all suggestions made by the reviewer and have implemented the necessary changes as noted below.

I think the paper needs to be a bit more careful in its use of terminology concerning several key concepts: blur, boundary and contrast. On line 18, for example we read that blurred edges "exhibit a spatial gradient of contrast orthogonal to the boundary contour". This is not quite accurate though, because such edges have a spatial gradient of luminance (not contrast) across the boundary contour. Spatial luminance contrast (a normalized measure of luminance difference in an image or across a contour) is a consequence of such luminance gradients. A gradient of contrast can indeed arise at texture boundaries for example, but these are quite different and have often been called '2nd order edges'. Similar comments apply to Line 39 where we read that blur is "the gradient of boundary contrast". Actually, both increasing blur, and decreasing contrast, will decrease the luminance gradient at an edge, and so the luminance gradient alone is not a good measure of blur (because the gradient also varies with contrast when blur is held constant).

We agree that 'contrast gradient' is awkward for the reasons correctly noted by the reviewer, and have settled on 'intensity gradient' to refer to the gradient of figure-ground stimulus intensity. We prefer to use intensity as opposed to luminance because for some neurons stimuli were presented at an equal luminance to the background, differing only in foreground chromaticity: we have added lines 18-21 to better describe the terminology and address this concern. Line 42 (previously Line 39) has also been appropriately revised according to the reviewer's suggestion.

Further, on line 45-46, we have "Blurring a stimulus boundary, as implemented here ... broadens the gradient of contrast across the shape's edges and thus decreases effective boundary contrast." Two comments: (i) as before, it's really the gradient of luminance that we're concerned with, and (ii) it's unclear why broadening the luminance gradient should be thought to decrease contrast. Yes, the local gradients are shallower, but if contrast is determined by overall luminance difference across the edge, this does not necessarily decrease when an isolated single edge is blurred; the full contrast may exist at a coarser spatial scale, if neighbouring edges are sufficiently far away. (It's true that when multiple edges are present, then as blur increases the edges start to merge or interact and contrast does go down, but the authors don't seem to have that in mind at this point.) [It's also true that a global measure of contrast, such as RMS contrast, is reduced by blur, but that also seems less relevant here since the authors are here talking about 'boundary contrast'.] My point is simply that terminology matters, and to help the reader we should try to get it right.

We agree with reviewer's comment and have now edited the relevant sentence (see lines 48-51) as follows: "Blurring a stimulus boundary, as implemented here, broadens the intensity gradient across a shape's boundary. Because the responses of roughly 80% of V4 neurons increase as figure-ground stimulus contrast is increased, one may expect blur to reduce the response of shape-selective neurons as edge intensity gradients are broadened."

Line 26: "neural mechanisms that underlie the computation and representation of blur remain unclear." Yes, that's true with respect to direct neurophysiological evidence, but computational models for perception and discrimination of blur, inspired by V1 physiology, are quite well developed, and might be worth citing (eg Watt & Morgan, Vision Res, 1985; Georgeson et al, Journal of Vision 2007) in addition to the work of Watson & Ahumada (2011) already cited.

We thank the reviewer for this important point. We have revised the relevant sentence and the suggested literature is now cited.

Lines 107-113 are intended to motivate a control experiment, but I thought the rationale here was not expressed clearly. For example, it's unclear what being different 'with respect to an 8-bit stimulus gamut' might mean. And it begins with (what seems to me) a rather unlikely hypothesis about what defines a luminance boundary: "that boundaries are identified from the contiguous region of a certain stimulus intensity". [It seems unlikely for many reasons, but to take just one: adding a modest amount of 2-D masking noise to one of these shapes would have little effect on boundary perception, but would completely eliminate the existence of any such 'contiguous region of a certain stimulus intensity', because every pixel would then be different from every other.]. Perhaps I've misunderstood something, but any clarification would be welcome. Lines 112-113 are clearer than what precedes them.

Re line 115, Fig 5 rather adds to the confusion because panel a implies that the shapes were white on a black background, while panel b shows the reverse (dark figures, white background). The Method section later reveals that both negative & positive contrasts were used, but it would be useful to clarify this with a few words in the main text or figure 5 legend. On the other hand, description of the results (lines 114-129) is clear and easily grasped.

We apologize for the lack of clarity in our contrast control experiment. Our goal was simply to say that level set contours determining stimulus area were defined from one 'grey-level' above background intensity, with background being 0 and stimulus foreground being 255 (one interval on an 8-bit representation). We have rewritten the offending section to make this clear: stimulus foreground is calculated as the subset of pixels distinct from the background as represented by a 24-bit RGB color display. Further, the reviewer's intuition regarding our analysis is correct: shape perception relying on a contiguous region of stimulus intensity is an implausible neural mechanism for boundary identification. However, this model is used only to calculate stimulus area under blur, and does not imply the visual system uses a similar physiological processes. Thus, we believe that our method of controlling for stimulus intensity covers the 'worst case scenario' for an intensity-based explanation of blur selectivity. We have adapted the description of the control analysis and the mentioned lines to make this argument clear (see lines 112-115 and 119-121). Our revisions also address the the positive vs. negative contrast concern evoked following Figure 5b.

*Line 171-172: "We then augment the APC model to include a blur-selective term, taken to be log-normal in blur factor beta, that scales shape-selective responses." This is a crucial statement of the APCB model favoured by the authors & implied by the data. It's fine as far as it goes, but it might help the reader for this statement to be reinforced by the sort of simple algebraic relation given on Line 184 for the not-favoured APC+B model. Presumably the APCB model states that cell response R to a given shape stimulus is the product of two terms, $R = APC(\cdot) * B(\cdot)$, where $APC(\cdot)$ is a function of stimulus shape but not blur, while $B(\cdot)$ is a function of the stimulus blur level beta, but independent of shape? If I've grasped this correctly, I think it would help the reader greatly to insert such a statement on line 172. It's quite hard to deduce this from the technical methods*

section, so if I've got it wrong, it would still be good to insert some correct version into the main text.

The reviewer is correct in their interpretation of the APCB model, where blur tuning multiplicatively scales shape tuning in a separable fashion. We have added a line to highlight this point (see lines 177-180).

Fig 8 and its associated text are good and important, but also hard work - perhaps necessarily so, because we are dealing with responses that exhibit a 4-way interaction between time, blur level, stimulus shape, and class of cell (blur-selective or not), with 3 output measures (spikes/s, blur modulation, and response latency). Multivariate, high-order interactions pretty much tax the human mind beyond its limits, so anything that could be added to further clarify or simplify, at least in part, would be welcome. (I know this is quite hard to do, without distorting the truth.

Figs 8e,f: x-axis is labelled as "Stimulus onset time (ms)", but I think it really means "Time after stimulus onset (ms)", and would be clearer if changed.

We have revised the description of Figure 8 to be less taxing on the reader and adopted the suggested axis label correction. Primarily, our revisions have removed certain unnecessary and overly-detailed phrases to improve readability without sacrificing accuracy.

Lines 234-240 are a supremely clear & succinct summary of the paper

Lines 255-6: "defocus, due to a finite depth of field or improper accommodation, may introduce optical (Gaussian) blur within a scene;" Delete Gaussian? I don't think that optical blur of this kind is actually Gaussian (though one might approximate it experimentally with some level Gaussian blur, as implied in line 257).

We thank the reviewer and agree that perceived focal blur of the human visual system is not necessarily Gaussian. We have revised that sentence (lines 267-269) accordingly.

Lines 282-3: "One candidate blur-selective circuit consists of a simple difference of SF power within V1...". Difference of power between what and what ? I think the reader needs to know this, in order to understand what 'This blur signal' is a couple of lines later.

Our intention was to describe a circuit wherein an excitatory response to intermediate spatial frequencies are inhibited by responses selective for high spatial frequencies, *i.e.* a difference of spatial frequency activity where high SF responses are subtracted from intermediate SF responses (see lines 313-316). Such a signal would be tuned for intermediate spatial frequencies, which is what we refer to as the 'blur signal'. We have revised the text to make this point clear.

Discussion section is generally very relevant, interesting, and thought-provoking.

We greatly appreciate this comment.

Reviewers' comments:

Reviewer #1 (Remarks to the Author):

The authors have done a great job introducing new analyses and addressing the limitations of the SRF approach in explaining their data. However, a big concern about the potential for well documented spatial frequency interactions ,which have been found in every area providing input to V4, for explaining these results remains. This goes to the crux of significance and interpretation: are these findings saying something special about what V4 does, or are they the somewhat inevitable consequence of already demonstrated receptive field interactions in earlier areas?

1. The authors largely do a good response of dealing with the SRF, in that its rotation insensitivity is inconsistent with observed shape selectivity. But the bigger issue of whether blur is even discernable from SF filtering is still problematic. As mentioned previously SF has a long history of psychophysical and physiological study and is nicely separable from things like orientation (unlike blur, which is why the authors have gone to such great lengths to make arguments about contrast, shape, size not being confounds). Notable there are relevant physiological studies, that would seem to be able to explain everything observed here, that are largely ignored. Around line 314, they suggest a spectral difference model that might explain their results, but suggest (at least in the response) that is a completely novel model and concept for dealing with SF. In fact, since at least 1992 (and probably earlier), we've known that the superimposition of higher SF, which by itself fails to excite primary visual cortex neurons, can powerfully suppress responses evoked by a preferred SF, and that this can create interesting SF tuning dynamics. (DeAngelis, Robson, Ohzawa, Freeman, 1992; Mazer, Vinje, McDermott, Gallant, 2002; Bredfeldt and Ringach, 2002; Ninomiya, Sanada, Ohzawa, 2012). Indeed, this likely to have origins or contributions from the LGN (Nolt, Kumbhani, Palmer, 2007). This seems completely sufficient to create the seemingly mysterious intermediate-blur neurons: without blur, high SF is present which is suppressing responses, and with blur, high SF is preferentially reduced, and the suppression is less. While some of the studies have talked about this in the context of V1 surround suppression (e.g., Webb, 2005, which would be perfectly fine for V4 sized RFs), many of these papers actually spatially superimpose the two frequencies within the classically defined RF. A broadly tuned suppression is commonly evoked with regard to normalization models, which have obvious computational advantages and have been well-explored. So, can the findings here simply be explained by normalization models, and the presence of a few neurons with preferred SFs aligned for the content of the shapes being used? Alternatively, if I did a V1 version of this experiment and blurred bars, given all the cited literature above, do the authors believe there would be no neurons with intermediate blur tuning? And, if I did LGN version with spots? Would it be fair under those circumstances to say that the LGN is computing or encoding blur?

2. The manuscript maintains that blur is a fundamental feature of visual scenes and suggests that it would be enormously advantageous to have "blur computations." But the authors already state that V1 populations implicitly code blur, and, given the papers cited above, these populations also include intermediate-blur neurons, which seems to be the main argument for an explicit code in this study. The manuscript and response letter suggest that this is on par with "face selectivity," something which cannot be simply explained by the RF of earlier areas and which is of clear ecological significance. But the fact that a small fraction of neurons have intermediate-blur selectivity (which would seemingly be explained by documented SF interactions) is not sufficient evidence for that claim. For example, many decades ago Hubel and Wiesel suggested that end-stopping could be useful for curvature detection. That's certainly true, but it's also useful for size selectivity and figure-ground segregation, so it's more appropriate to say that the representations in V1 could support curvative detection than it is to say curvature analysis or detection is "occurring in" or "the purpose of" V1.

3. One interesting point that could be elaborated upon in the discussion is related to the observation, demonstrated by the separable shape-blur model, that blur is NOT affecting shape selectivity. I think this could be "stated" as a type of invariance that is very ecologically useful, given there are many conditions (such as presbyopia) that cause blur.

4. As is clear by the number of experiments and analyses, there are lots of potential confounds when dealing with blur. The authors have done an admirable job with most of these, but I was less than convinced by the size/blur analyses, in which the failure to find significant additive firing rate interactions, is taken as evidence that changes in size cannot be responsible for blur-related changes in responsiveness. As the experimental design makes clear, one could argue that blur is either increasing or decreasing the size (depending on the luminance or contrast sensitivity non-linearities), and it's not clear which it would be, or for that matter, why a 10% change in size is a reasonable guess for the effects of blur. Is there a reason why a modeling approach, such as is down with shape and blur, wouldn't work?

Reviewer #2 (Remarks to the Author):

The authors addressed all my concerns. I appreciate the addition of data in Figure S1.

Reviewer #3 (Remarks to the Author):

I have carefully considered the authors' response letter for this revised manuscript, and read through the revised paper. I'm satisfied that the authors have dealt with my comments very appropriately. The revisions made in light of other reviewers' comments seemed to me to be fair and reasonable as well.

I think the paper is now ready for publication.

Reviewer # 1

We are pleased to have satisfied the concerns of Reviewers #2 and #3, and address the outstanding issues noted by Reviewer #1.

The authors have done a great job introducing new analyses and addressing the limitations of the SRF approach in explaining their data. However, a big concern about the potential for well documented spatial frequency interactions ,which have been found in every area providing input to V4, for explaining these results remains. This goes to the crux of significance and interpretation: are these findings saying something special about what V4 does, or are they the somewhat inevitable consequence of already demonstrated receptive field interactions in earlier areas?

Our study provides the first documentation of the simultaneous representation of shape and blur in primate visual cortex. The significance of our finding lies in the novelty of this demonstration. We do not believe that our results are an inevitable consequence of previously demonstrated tuning interactions in earlier areas.

Critically, we demonstrate that V4 neurons exhibit tuning for boundary form *and* blur. As noted by the reviewer and described in our manuscript’s Discussion section, selectivity for intermediate blur could arise from SF tuning interactions from earlier visual areas. However, since tuning for boundary form cannot be explained on the basis of SF power in visual stimuli (Oleskiw *et al.*, 2014), our results cannot be explained by SF interactions performed by neurons of earlier visual areas alone. The results reported here suggest that responses of many V4 neurons reflect two distinct mechanisms: the encoding of boundary conformation, based on localized orientation in space, and encoding of blur, based on band-selective spatial frequency information. A joint model of shape and blur coding, as proposed in our study, is required to explain our results.

1. The authors largely do a good response of dealing with the SRF, in that its rotation insensitivity is inconsistent with observed shape selectivity. But the bigger issue of whether blur is even discernable from SF filtering is still problematic. As mentioned previously SF has a long history of psychophysical and physiological study and is nicely separable from things like orientation (unlike blur, which is why the authors have gone to such great lengths to make arguments about contrast, shape, size not being confounds). Notable there are relevant physiological studies, that would seem to be able to explain everything observed here, that are largely ignored. Around line 314, they suggest a spectral difference model that might explain their results, but suggest (at least in the response) that is a completely novel model and concept for dealing with SF. In fact, since at least 1992 (and probably earlier), we’ve known that the superimposition of higher SF, which by itself fails to excite primary visual cortex neurons, can powerfully suppress responses evoked by a preferred SF, and that this can create interesting SF tuning dynamics. (DeAngelis, Robson, Ohzawa, Freeman, 1992; Mazer, Vinje, McDermott, Gallant, 2002; Bredfeldt and Ringach, 2002; Ninomiya, Sanada, Ohzawa, 2012). Indeed, this likely to have origins or contributions from the LGN (Nolt, Kumbhani, Palmer, 2007). This seems completely sufficient to create the seemingly mysterious intermediate-blur neurons: without blur, high SF is present which is suppressing responses, and with blur, high SF is preferentially reduced, and the suppression is less. While some of the studies have talked about this in the context of V1 surround suppression (e.g., Webb, 2005, which would perfectly fine for V4 sized RFs), many of these papers actually spatially superimpose the two frequencies within the classically defined RF. A broadly tuned suppression is commonly evoked with regard to normalization models, which have obvious computational advantages and have been well-explored.

We thank the reviewer for providing these relevant references. In our revised manuscript we expand our discussion of how suppression of neuronal responses due to the presence of high SF in the visual stimulus could give rise to a preference for intermediate-blur (see lines 323-326). However, the extant literature characterizing SF interactions in the earlier areas do not directly predict our results for numerous reasons:

(i) There is a lot of diversity across studies in terms of the SF interactions documented. While some studies report suppression from high SF, others report suppression at low SF and still others report broadband SF suppression. Some studies report that SF modulation could depend on the relative phase of each grating (De Valois & Tootell, 1983). There is also conflict between results from different studies (*e.g.* compare Ninomiya *et al.* to Bredfeldt and Ringach) and possibly between the cat and monkey. Thus, no coherent picture emerges from the available literature about SF interactions in earlier areas. We are also not aware of any published report that proposes a model wherein suppression from high SF content gives rise to tuning for intermediate blur stimuli. Thus, we do not believe that any reader could simply look at the conflicted literature of SF interactions and immediately predict the results of our study. Instead, after reading our result, an astute reader may piece together how SF interactions in V1 *could* give rise to intermediate blur tuning, as Reviewer #1 does, but this in no way diminishes the significance of our findings.

(ii) SF interactions in earlier areas have largely been discussed in the context of cross-orientation suppression and the temporal dynamics of SF tuning. To the extent we are aware, such computations have not been discussed in the context of blur processing in natural vision. Thus, our study is significant in providing a potential explanation for why earlier visual areas, *i.e.* LGN and V1, might perform such SF computations, namely toward encoding the pervasive attribute of blur in naturalistic scenes.

(iii) Not all V4 neurons are tuned to intermediate blur, *e.g.* cells a08 and b29 of Fig 2b,c. Thus, while high-SF suppression could contribute to the responses of intermediate blur tuned neurons, they cannot explain the responses of other V4 neurons in our study.

So, can the findings here simply be explained by normalization models, and the presence of a few neurons with preferred SFs aligned for the content of the shapes being used? Alternatively, if I did a V1 version of this experiment and blurred bars, given all the cited literature above, do the authors believe there would be no neurons with intermediate blur tuning? And, if I did LGN version with spots? Would it be fair under those circumstances to say that the LGN is computing or encoding blur?

As discussed in our manuscript, normalization could *contribute* to the observed selectivity for intermediate blur in V4. Our results suggest that blur tuning multiplicatively scales shape selectivity in V4, and we have proposed a simple yet novel model to explain these findings. We believe this concept was not adequately conveyed in our discussion, and we have revised this important point to make it clear (see lines 331-333). We thank the reviewer for raising this concern.

We are in no way opposed to the idea that the LGN and V1 (and the retina) carry out computations that are important for downstream processing. Therefore, if V1 shows intermediate blur tuning to oriented edges, and if our study spurs this realization and connects these two findings, our study’s publication would have additional value. We have not presented such blurred stimuli in earlier visual areas, but completely agree that it would be useful to do so.

Finally, since selectivity for boundary conformation has not been demonstrated in LGN or V1, we would predict that LGN/V1 responses to the blurred shape stimuli used in our study would be distinct from V4 responses in that they would not jointly encode blur and shape information.

2. The manuscript maintains that blur is a fundamental feature of visual scenes and suggests that it would be enormously advantageous to have “blur computations.” But the authors already state that V1 populations implicitly code blur, and, given the papers cited above, these populations also include intermediate-blur neurons, which seems to

be the main argument for an explicit code in this study. The manuscript and response letter suggest that this on par with “face selectivity,” something which cannot be simply explained by the RF of earlier areas and which is of clear ecological significance. But the fact that a small fraction of neurons have intermediate-blur selectivity (which would seemingly be explained by documented SF interactions) is not sufficient evidence for that claim. For example, many decades ago Hubel and Wiesel suggested that end-stopping could be useful for curvature detection. That’s certainly true, but it’s also useful for size selectivity and figure-ground segregation, so it’s more appropriate to say that the representations in V1 could support curvative detection than it is to say curvature analysis or detection is “occurring in” or “the purpose of” V1.

Joint coding of shape and blur cannot be explained simply by SF interactions within earlier areas. As there is no explicit representation of boundary shape in regions projecting to V4, our results suggest that the neural code for boundary conformation and blur first arises in V4, directly analogous to how face selectivity emerges from earlier visual areas without an explicit representation of faces. While a simple representation of blur may indeed be achieved in V1 alone, we propose that the joint encoding of shape and blur in V4 is an explicit computation of enormous ecological relevance not previously reported in the literature.

3. One interesting point that could be elaborated upon in the discussion is related to the observation, demonstrated by the separable shape-blur model, that blur is NOT affecting shape selectivity. I think this could be “stated” as a type of invariance that is very ecologically useful, given there are many conditions (such as presbyopia) that cause blur.

We thank the reviewer for noting this important point. We have added an additional discussion statement to highlight this concept and note the ecological utility of such cells (see lines 258-260).

4. As is clear by the number of experiments and analyses, there are lots of potential confounds when dealing with blur. The authors have done an admirable job with most of these, but I was less than convinced by the size/blur analyses, in which the failure to find significant additive firing rate interactions, is taken as evidence that changes in size cannot be responsible for blur-related changes in responsiveness. As the experimental design makes clear, one could argue that blur is either increasing or decreasing the size (depending on the luminance or contrast sensitivity non-linearities), and it’s not clear which it would be, or for that matter, why a 10% change in size is a reasonable guess for the effects of blur. Is there a reason why a modeling approach, such as is down with shape and blur, wouldn’t work?

We thanks the reviewer for suggesting the additional modelling analysis of our size control data and have implemented two additional analyses to investigate the hypothesis that a size confound could explain our data. Results are consistent with all previous findings.

In our first analysis we asked if shape selectivity could explain size control data, since local curvatures change as a stimulus is scaled. Here, level-set contours were computed from scaled and blurred stimuli, as was done in our control experiment for curvature modification. For each neuron, an APC model fit to shape data was used to predict size control data. We found these models poorly explained size control data, indicating size confounds to be incompatible with a neuron’s tuning for boundary conformation.

In our second analysis we asked if response modulation was better explained by blur or size, *i.e.* comparing performance of an $APC * B$ model to an $APC * S$ model, where S is a Gaussian tuning function for size that multiplicatively scales a neuron’s selectivity for shape. Stimulus size was determined from the arc length of level-set contours computed from scaled and blurred size control data. We found that neurons were overwhelmingly better fit by the $APC * B$ model than the

*APC * S* model, indicating blur to explains more variance of our data, despite both models having identical number of free parameters and functional form.

We now summarize these additional analyses of our size control experiment in Results (see lines 109-113 and 422-434). Should they agree, we would like to acknowledge Reviewer #1 in our manuscript for suggesting these experiments as they have significantly strengthened our original claims.

REVIEWERS' COMMENTS:

Reviewer #1 (Remarks to the Author):

I agree with the authors that the SF-interaction story is only a possible explanation, but I wanted to make sure the manuscript brought it up (because it seems like a particularly parsimonious explanation to me).

And the size analysis is also improved. I recommend publication: it's an interesting data set and the analyses are very sophisticated and appropriate.

Final remarks to the authors

We are pleased to have satisfied the concerns of all reviewers:

Reviewer #1

I agree with the authors that the SF-interaction story is only a possible explanation, but I wanted to make sure the manuscript brought it up (because it seems like a particularly parsimonious explanation to me).

And the size analysis is also improved. I recommend publication: it's an interesting data set and the analyses are very sophisticated and appropriate.

Reviewer #2

The authors addressed all my concerns. I appreciate the addition of data in Figure S1.

Reviewer #3

I have carefully considered the authors' response letter for this revised manuscript, and read through the revised paper. I'm satisfied that the authors have dealt with my comments very appropriately. The revisions made in light of other reviewers' comments seemed to me to be fair and reasonable as well.

I think the paper is now ready for publication.